# Imaging and Genetic Tools for the Investigation of the Endocannabinoid System in the CNS

**DOI:** 10.3390/ijms242115829

**Published:** 2023-10-31

**Authors:** Armin Kouchaeknejad, Gunter Van Der Walt, Maria Helena De Donato, Emma Puighermanal

**Affiliations:** Neuroscience Institute, Autonomous University of Barcelona, 08193 Bellaterra, Spain; armin.kouchaeknejad@uab.cat (A.K.); gunter.vanderwalt@uab.cat (G.V.D.W.); mariahelena.dedonato@autonoma.cat (M.H.D.D.)

**Keywords:** endocannabinoids, CB1R, genetic mouse models, imaging tools, CNS

## Abstract

As central nervous system (CNS)-related disorders present an increasing cause of global morbidity, mortality, and high pressure on our healthcare system, there is an urgent need for new insights and treatment options. The endocannabinoid system (ECS) is a critical network of endogenous compounds, receptors, and enzymes that contribute to CNS development and regulation. Given its multifaceted involvement in neurobiology and its significance in various CNS disorders, the ECS as a whole is considered a promising therapeutic target. Despite significant advances in our understanding of the ECS’s role in the CNS, its complex architecture and extensive crosstalk with other biological systems present challenges for research and clinical advancements. To bridge these knowledge gaps and unlock the full therapeutic potential of ECS interventions in CNS-related disorders, a plethora of molecular–genetic tools have been developed in recent years. Here, we review some of the most impactful tools for investigating the neurological aspects of the ECS. We first provide a brief introduction to the ECS components, including cannabinoid receptors, endocannabinoids, and metabolic enzymes, emphasizing their complexity. This is followed by an exploration of cutting-edge imaging tools and genetic models aimed at elucidating the roles of these principal ECS components. Special emphasis is placed on their relevance in the context of CNS and its associated disorders.

## 1. The Endocannabinoid System (ECS)

Among the wide variety of biomolecules that mediate neuronal biology, lipids have critical structural, signalling, and metabolic functions in the CNS [1,2]. The ECS is a widespread lipid-based, neuromodulatory network with critical roles in regulating many physiological processes in the CNS. The ECS directly influences emotional behaviour, appetite, reward, memory, learning, and pain sensations, among others. Furthermore, the ECS regulates the CNS during its development and in adulthood by modulating homeostasis, synaptic plasticity, neurogenesis, and responses to endogenous and environmental insults [3,4,5].

The ECS is comprised of endogenous lipid signalling molecules, better known as endocannabinoids (eCBs), cannabinoid receptors, and all the enzymes responsible for eCB metabolism (Table 1). Recent studies also expand this definition to include key proteins that mediate systemic, transmembrane, and intracellular transport of eCBs [6]. Following the multiple and varied roles of the ECS in the CNS, its involvement in several CNS-related diseases from cancer to neurodegenerative disorders is frequently highlighted. A growing body of literature emphasises how imaging tools and genetical manipulations of key ECS components may offer promising insights and therapeutic opportunities in CNS disorders [7,8]. In the following section, we briefly introduce the different ECS components, before reviewing these tools.

### 1.1. Cannabinoid Receptors

Endocannabinoids can bind a wide variety of receptors, including the core cannabinoid receptor type 1 and 2 (CB1R and CB2R) as well as others such as G-protein-coupled receptors (GPR55, GPR18, GPR119), transient receptor potential vanilloid type 1 (TRPV1), and peroxisome proliferator-activated receptors (PPARα and PPARγ) (Table 1).

CB1R and CB2R are considered to be the main target for endogenous, plant-derived, and synthetic cannabinoids and are thus the best characterised receptors of the ECS [9]. As G-protein-coupled receptors (GPCRs), their activation triggers different effectors such as inwardly rectifying potassium channels (GIRKs), arrestins, adenylyl cyclase, mitogen-activated protein kinases, and some voltage-dependent calcium channels [7]. As a result, CBR activation can regulate a wide variety of cellular functions such as transcription, synaptic plasticity, and general cell physiology [10].

CB1R in humans is a 472-amino-acid protein, encoded by the *CNR1* gene. It is highly expressed on axon segments and terminals of interneurons and glutamatergic, cholinergic, glycinergic, and serotonergic neurons [11]. Furthermore, CB1R distribution varies between brain regions such as the cortex, hippocampus, basal ganglia, and cerebellum. Recently, CB1R has been shown to be associated with subcellular organelles such as mitochondria [12]. CB2R is encoded by *CNR2* in humans, with a full length of 360 amino acids. It is far less abundant than CB1R in the brain and is mainly found in microglia and vascular elements of the CNS, but substantial increases in CB2R expression can be induced with injury or inflammation [7,13,14].

The other receptors are not as well characterised as CB1R and CB2R and are rather considered as part of the extended ECS, or endocannabinoidome. However, CB1R/CB2R alone fail to explain the totality of eCB functions [15] and therefore the influence of other related receptors deserves further study. TRPV1 is a TRP channel activated by the eCB anandamide (*N*-arachidonoylethanolamine; AEA), among other endogenous ligands. It is regarded as an ionotropic CBR due to increasing evidence for its interaction with cannabinoids. This receptor often colocalizes with CB1R, suggesting a simultaneous activation by AEA. Interestingly, these two receptors mediate diametrically opposite effects on neuronal activity: TRPV1 tends to increase activity by regulating calcineurin and ATM serine–threonine kinase and promoting cation influx, while CB1R frequently inhibits synaptic activity. This contrast may contribute to the inconsistent role of AEA in anxiety and fear studies. Other receptors from the TRP channel family have been reported to bind cannabinoids. For example, AEA can bind to TRPM8, while the main psychoactive constituent of *Cannabis sativa*, Δ^9^-tetrahydrocannabinol (THC), can exert physiological actions through TRPV2, TRPV3, and TRPV4 binding. However, current knowledge on their mechanisms and roles in pathophysiology is scarce, necessitating further studies to understand the importance of TRP channels in the ECS [16,17].

The nuclear PPARα, PPARγ, and PPARδ receptors are also reported to be cannabinoid targets, with potentially a significant influence on gene transcription [18]. However, despite recent evidence implicating PPARs’ involvement in cannabinoid-induced neuroprotection and analgesia [8], the role of PPARs as true CBRs is still disputed.

Other GPCRs such as GPR55, GPR18, and GPR119, are also reportedly regulated by endocannabinoids but their interactions require more extensive study [19,20]. In particular, GPR55 can bind cannabinoids and some researchers designate this receptor as a putative CB3R [21]. However, GPR55 also interacts with non-cannabinoid ligands, shares a low amino acid homology with CB1R/CB2R, and interacts with different signalling pathways involving Rho-associated protein kinase (ROCK); thus, its designation as a true CBR is still disputed. GPR18 presents some sequence similarities to GPR55 in critical positions and binds a wide range of eCBs mediating therapeutic effects. Similarly, however, the GPR18 pharmacology and mechanism of action have yet to be clarified [22]. Therefore, this review will focus on CB1R and CB2R.

#### Intracellular Organelle-Associated CB1R

Like the majority of GPCRs, CB1R is mainly localized to the plasma membrane. However, recent evidence suggests CB1R can also associate with organelles such as the endoplasmic reticulum, lysosomes, and mitochondria [14,23]. Specific focus has been placed on the CB1R associated with mitochondrial membranes (mtCB1R) and its signalling properties that reportedly differ from their cell-surface-localized counterparts [12]. Hebert-Chatelain et al. have demonstrated how the activation of G proteins coupled to the putative mtCB1R receptors in hippocampal neurons could offer an explanation for cannabinoid-induced amnesia [24]. Briefly, mtCB1R activation led to a G-protein-dependent inhibition of soluble-adenylyl cyclase, subsequently decreasing PKA-dependent phosphorylation of NDUFS2, a subunit of the complex I electron transport chain involved in mitochondrial respiration. This led to a decrease in cellular respiration, eventually affecting memory formation in mice exposed to cannabinoids.

Jimenez-Blasco et al. reported similar effects on complex I but related them to mtCB1R present in astrocytes [25]. Skupio et al. proposed that mtCB1R modulates calcium levels in brain circuits, regulating the effects of stress and cortisone in mice tested for novel object recognition [26]. Another investigation linking mtCB1R to cannabinoid effects was conducted by Soria-Gomez et al., showing that mtCB1R decreases synaptic transmission and induces catalepsy [27]. Overall, significant progress related to mtCB1R has been made over the last few years; however, its role in CNS physiology warrants further study.

### 1.2. Endocannabinoids: Endogenous CBR Ligands

Endocannabinoids (eCBs) are the endogenous lipid signalling molecules that exert their physiological effects by interacting with CBRs. They are often contrasted and compared to plant-derived (phyto-) cannabinoids such as THC, which exerts most of its effects through CB1R agonism. 2-Arachidonoylglycerol (2-AG) and AEA are the first-described and best characterised eCBs, and thought to mediate the majority of ECS function [28,29].

AEA was discovered first in 1992, followed by 2-AG in 1995, while researchers were searching for the endogenous ligands of CBRs. Both compounds are key autocrine and paracrine second messengers derived from phospholipid metabolism [19,30].

While almost all neurotransmitters are pre-synthesized and stored in vesicles until their release, eCBs seem to be unique in that they are produced “on demand” at the synapse to modulate synaptic activity. This is mainly derived from the observation that the first steps in eCB biosynthesis are mediated by enzymes activated during membrane depolarization upon neurotransmitter binding [29]. However, the “on demand” release model remains controversial due to the lack of clarity regarding the molecular mechanisms involved and virtual inability to experimentally distinguish between on-demand synthesis and canonical vesicular release.

All eCBs are enzymatically synthesized from abundant membrane phospholipids, predominantly conjugated with polyunsaturated fatty acids (PUFAs). Specifically, 2-AG is majorly derived from membrane phosphatidylinositols (PIs), substituted with arachidonic acid, while AEA is synthesized from N-arachidonoyl phosphatidylethanolamine (PE) [3,31]. Between these two major eCBs, 2-AG is much more abundant and has a significantly higher agonist efficacy for both CB1R and CB2R than AEA [13]. Even though 2-AG and AEA are considered as the major eCBs, many of their congeners from the N-acylethanolamine (NAE) and monoacylglycerol (MAG) signalling lipid classes are also thought to interact with the ECS (Table 1; Figure 1) [32]. These molecules include 2-O-arachidonoylethanolamine (virodhamine or O-AEA), N-docosahexaenoylethanolamine (DHEA), N-eicosapentaenoylethanolamine (EPEA), N-palmitoylethanolamine (PEA), N-oleoylethanolamine (OEA), N-stearoylethanolamine (SEA), docosatetraenoylethanolamide (DEA), N-linoleoylethanolamine (LEA), dihomo-γ-linolenoyl-ethanolamide (DGLEA), arachidonyl glycerol ether (noladin ether or 2-AGE), and 2-oleoylglycerol (2-OG). The discovery of two endogenous CB1R agonists, N-arachidonoyldopamine (NADA) and 2-arachidonoyl lysophosphatidylinositol (2-ALPI), that are not NAEs or MAGs implies that many other bioactive lipids with a similar structure may also act as eCBs. These compounds are still poorly studied in comparison to 2-AG and AEA in the context of the ECS and CNS; thus, this review will focus on 2-AG and AEA.

### 1.3. The Enzymes

One of the main features of eCBs is that they are likely synthetized “on demand” with precise spatiotemporal coordination. This model implies that the differential distribution of eCB biosynthetic pathways in different brain areas, cell types, and physiological states would dictate their effect in that specific milieu [32,33,34]. For instance, AEA metabolism relies on different sets of enzymes with differential distribution throughout the CNS than those that mediate 2-AG metabolism [35]. Another prime example of this functional diversity is the differential roles of neuronal and glial ECS function in specific CNS contexts. To date, astrocytic CB1R and 2-AG catabolic enzymes have been shown to have differential effects on synaptic plasticity, hippocampal memory processing, and cerebellar function than their neuronal counterparts [36,37,38,39]. KO of these components produced differential effects dependent on the neuronal layers and cell types involved, emphasizing the effect of differential metabolic component expression in different cellular contexts. Furthermore, evidence suggests active shuttling of AA and 2-AG between neurons and astrocytes to drive eCB metabolism and function [40], while microglial 2-AG synthesis is regulated separately by a different catabolic enzyme [33].

Arachidonic acid and other PUFAs are the precursors to both eCBs and eicosanoids, a class of oxidized lipid species involved in diverse cellular functions from growth to immune responses (e.g., prostaglandins, thromboxanes, and lipoxins). Due to this structural similarity between eCBs and eicosanoids, the enzymes responsible for eCB metabolism are also those responsible for the synthesis and catabolism of other eicosanoids.

2-AG is mainly produced with a two-step, sequential hydrolysis reaction. This first involves the conversion of phosphatidylinositol bisphosphate to diacylglycerol (DAG), catalysed by phospholipase C (PLC). Thereafter, a diacylglycerol lipase (DAGL) catalyses removal of an acyl group from DAG to produce 2-AG (Figure 2). The latter has been extensively studied as a therapeutic target and presents two abundant isoforms in the brain (DAGLα and DAGLβ). Alternative pathways for 2-AG synthesis have been suggested, such as the sequential cleavage of a phosphatidyl inositol precursor by a phospholipase A (PLA_1_) and lyso-PLC. However, the precise mechanisms and significance of these alternative pathways in the CNS have yet to be fully characterised [13]. 2-AG degradation is mainly catalysed by monoacylglycerol lipase (MAGL), producing arachidonate and glycerol, or by the alpha/beta domain hydrolases (ABHD6/12) to form the same products. MAGL is primarily localized to presynaptic terminals, while ABDH6 is localized to excitatory dendrites, dendritic spines, and glial cells. Nevertheless, inhibition of either enzyme increases 2-AG and CB1R-mediated signalling [41]. The role of ABDH12 has not been clarified to date. The specific localization of 2-AG metabolic enzymes is in line with what would be expected given the core roles of eCBs in retrograde synaptic regulation: where neurotransmitter binding triggers eCB synthesis in the postsynaptic neuron and it is degraded after interacting with presynaptic CBRs in the afferent axon.

AEA biosynthesis is thought to start from N-acylation of PE via enzymes including cytosolic PLA_2_ε. The produced N-arachidonoyl phosphatidyl ethanolamine (NAPE) then undergoes hydrolysis via a substrate-specific phospholipase D (NAPE-PLD) [7]. However, several other pathways from NAPE to AEA have been suggested, including (1) ABHD4 and glycerophosphodiesterase 1 (GDE1) with a glycerophospho-AEA intermediate, (2) PLA2 and ABHD4 with a lyso-AEA intermediate, (3) PLC, and (4) phosphatases such as PTN22 and SHIP1. In contrast, AEA degradation in the CNS is far less ambiguous. AEA catabolism is mainly catalysed by fatty acid amide hydrolase (FAAH) or N-acylethanolamine-hydrolysing amidase (NAAA) as a main alternative [42] (Figure 2). Both these eCBs and AA can also be catabolized by cyclooxygenase (COX-2) to produce bioactive prostaglandins, or by lipoxygenases (12/15-LO) and cytochrome P450 (CYP2-4) to produce other bioactive eicosanoids [29].

A significant and contentious topic regarding metabolism of eCBs is their transport between and within cells, as their receptor targets and metabolic effectors are spread throughout the intra- and intercellular space. Two competing hypotheses regarding eCB trafficking rely on two contrasting general mechanisms for small molecule transport across the plasma membrane. Molecules can either pass through the cell membrane unaided if they are sufficiently soluble in the hydrophobic phospholipid bilayer (simple diffusion). More complex and/or hydrophilic molecules often require a specific protein channel/transporter to mediate its transmembrane transport (facilitated transport) [6].

Briefly, the simple diffusion model states that the hydrophobic, lipidic eCBs can easily diffuse through the cell membrane. This is supported with non-saturable kinetics measured for eCB uptake and their effective uptake in vastly different cell systems, as expected for a process that does not require a specific protein transporter with a saturable binding domain. However, this hypothesis does not account for the significant distances these lipidic molecules must cross within the aqueous intra- and extracellular environment, specifically during synaptic trafficking.

The second hypothesis suggests a putative eCB membrane transporter (EMT) involved in facilitating diffusion of both AEA and 2-AG across the synapse and cell membrane (Figure 2). The main support for this theory comes from evidence that expected EMT inhibitors can increase brain eCB levels and enhance their cannabimimetic effects [19]. Furthermore, several proteins have been shown to participate in intracellular and possibly transmembrane eCB transport in the CNS [43,44]. Fatty acid binding proteins (FABP5/7), heat shock protein 70 (Hsp-70), sterol carrier proteins (Scp2/x), and FAAH variants such as FLAT were implicated, but the eCB transport functions of these proteins have yet to be validated in vivo. Strong recent evidence suggests vesicular exocytosis dependant on specific proteins may also be involved in eCB trafficking across the synapse. Both α-synuclein (a protein often affected in Parkinson’s disease [45]) and core vesicle proteins known as SNAREs have been shown to mediate eCB synaptic regulation by facilitating postsynaptic release of AEA and 2-AG [46]. Nevertheless, authors do not exclude the possibility that non-vesicular mechanisms may be involved as α-synuclein could also mediate direct eCB release or interact with other potential eCB carriers such as FABP5.

## 2. Imaging the ECS

Understanding the distribution and dynamics of ECS components is crucial for investigating its role in the CNS and related disorders, yet this is complicated with the heterogenous and interconnected nature of the system. Most of the ECS components are multifunctional, impacting and being influenced by numerous other signalling pathways. As a result, instead of being a distinct and isolated system, the ECS is extensively connected to several other homeostatic pathways that are associated with a wide array of functions and disorders. Examples include the enzymes shared between eCB metabolism and that of prostaglandins and thromboxanes that control inflammation [29], or the dimerization of CBRs with other neurotransmitter receptors [47,48,49]. To overcome these limitations, a wide range of imaging tools have been exploited to study the ECS. These range from positron emission tomography (PET), autoradiography, and conventional immunohistochemistry to cutting-edge tools such as ultra-resolution and electron microscopy and engineered sensors. Every technique presents its advantages and disadvantages; thus, the best tool for a given application is dependent on several factors. As we will expand in this section, the optimal choice depends on the desired resolution, antibody and equipment availability, tissue matrix and model system selection, etc. The aim of this section is to clearly summarize the most used and promising imaging techniques for probing the various ECS components. To this end, we present a brief general overview of the principal imaging tools available, citing the technical advantages and disadvantages (Table 2), before reviewing their application towards individual ECS components.

### 2.1. The Most Used Tools in ECS Imaging

Many initial studies on ECS proteins employed radioisotope-labelled ligands (radioligands) to investigate their tissue distribution and infer their pharmacological properties (Table 2). Radioligands are coupled with non-invasive techniques such as PET to obtain 3D organ images or ex vivo autoradiography or radioligand binding assays. Therefore, they present relevant alternatives to immunocytochemistry (ICC) or immunohistochemistry (IHC), especially in the absence of validated antibodies [50,51].

ICC and IHC are still some of the most widely used techniques to study the cellular and subcellular ECS organization [1,52]. Utilizing conventional fluorescence light microscopy, these tools facilitate visualization of specific target structures with high sensitivity and adequate resolution. The potential limitations of immunofluorescence include limited resolution, low tissue penetration, and potential interference by endogenous molecules that impede affinity and selectivity. Fluorescence resonance energy transfer (FRET) greatly improves the resolution of optical microscopy, creating a midway between ICC and expensive electron microscopes [52]. This technique exploits an energy transfer between two closely positioned fluorophores and is mainly used to determine the proximity of two or more biomolecules to each other. Thus, one of its main limitations is a high dependence on the target distance while it also lacks appropriate labelling for intracellular proteins.

The next tier in optical power comes with the electron microscope. Transmission electron microscopy (TEM) has been the most widely exploited tool in this category for ECS investigations. TEM has a very high resolution but, compared to optical microscopy, is more expensive, time-consuming, and involves laborious sample preparation with high risk of contamination [52].

More recent tools have enabled huge improvements in imaging resolution, giving access to information with nanoscale distribution and unprecedented detail. A notable example from recent ECS studies is Stochastic Optical Localization Microscopy (STORM). Specifically, this technique has high lateral, axial, and spatial resolution and can be used for 3D analyses, but requires specialized fluorophores and produces enormous datasets, which impedes rapid workflows [1,52].

Mass spectrometry imaging is the main technique used to image eCBs specifically and has high sensitivity, mass resolution, and accuracy, but has no spatiotemporal resolution and limited quantitative power. Recently, bioengineered sensors have overcome these limitations, providing high selectivity and sensitivity and favourable kinetics, but their potential is yet to be fully exploited.

### 2.2. CB1R Imaging

Radioactive ligands have been a primary and extensively used tool, especially for receptor imaging. Therefore, there are several radioactively tagged agonist and antagonist ligands available for imaging with PET, autoradiography, or radioligand binding. For in vivo or ex vivo CB1R visualization, studies have mainly used PET imaging with [^18^F]- or [^11^C]-labelled ligands [53] (Table 3). For example, Burns et al. were able to investigate the selective inverse agonist [^18^F]MK-9470 both with PET and autoradiography, confirming its potential as an imaging tool for CB1R. Similarly, DPR_211_ was exploited for PET scanning studies by Chang et al., showing significant enrichment in the brain [54,55]. Another interesting [^18^F]-labelled radioligand is FMPEP-d_2_ that has been used to assess CB1R distribution in Alzheimer’s disease mouse models, among other neurodegenerative diseases. [^11^C]-labelled radioligands offer the advantage of a faster synthesis, but they have a shorter half-life, which may limit quantification of radioligand kinetics. The most promising radioligands in this category are [^11^C]-OMAR and [^11^C]COCl_2_ [56,57].

Predictably, conventional fluorescence light microscopy has been extensively used for CB1R imaging, i.e., for elucidating its distribution from the organ to subcellular level with IHC. In the CNS, CB1R has been found in a wide range of regions (e.g., spinal cord, neocortex, olfactory bulb, striatum, amygdala, hippocampus, prefrontal cortex, cerebellum, and hypothalamus) and in different cell types ranging from glutamatergic neurons to astrocytes, microglia, and oligodendrocytes. For example, Tsou et al. generated a highly specific polyclonal antibody targeting the N-terminal CB1R to demonstrate its enriched distribution in the rat forebrain [58]. Egertová and Elphick analysed the expression of CB1R utilising C-terminal antibodies in rat brains, revealing the receptor’s subcellular locations [59]. Katona et al. assessed CB1R expression in the amygdala and elucidated its role in GABAergic synaptic transmission [60]. Navarrete et al. exploited rabbit polyclonal CB1R antibodies to demonstrate eCB-mediated neuron–astrocyte communication [61]. At the subcellular level, CB1R expression was shown at both the postsynaptic and presynaptic membranes as well as on mitochondrial-associated membranes and neuronal soma [24]. Marinelli and colleagues were able to confirm eCB-mediated regulation of interneurons and pyramidal cells using a similar strategy [62]. It should be noted that antibodies targeting the N- and C-terminal of CB1R have shown different effectivity for different applications. While an in-depth analysis thereof is outside the scope of this review, we refer the reader to the studies cited above for more detail.

Other types of probes have also been used for optical imaging. These are divided into covalent probes (i.e., photoactivable, electrophilic, and bifunctional) and fluorescent probes. Covalent probes for CB1R have mainly been used in affinity assays with relatively low application in imaging studies. Fluorescence probes, however, allow real-time assessment with a high spatiotemporal precision and have been used in imaging studies. Their downside is that a link between the pharmacophore and the fluorescent tag is required as the probe affects various physicochemical and photophysical properties of the final conjugate [53]. An interesting application of fluorescent probes was found by Martin-Couce et al. where a biotinylated 2-AGE analogue complex was successfully used to visualize CB1R in neurons [47].

FRET provided additional resolution on interactions between CBRs and other receptors. Many studies focused on the potential heteromerization of CB1R with orexin receptor 1 (OX-1R), considering their shared role in ERK1 and ERK2 activation and thus eCB signalling regulation. For example, Imperatore et al. demonstrated the formation of the OX-1R/CB1R heteromer [48]. Hojo et al. investigated another interaction between CB1R and the μ-opioid receptor (μOR), demonstrating the formation of a μOR/CB1R complex that transmits a signal though a common G protein [63].

Electron microscopy, particularly TEM, has been exploited in different investigations to provide a very high-resolution description of CB1R [64,65,66]. Nyíri et al. confirmed previous results on CB1R distribution, showing significant receptor density on the preterminal and terminal axonal segments in GABAergic and glutamatergic cells, as well as astrocytes [67]. Moreover, new tools such as immunogold imaging coupled with TEM provide even further opportunities. As previously stated, Benard et al. utilised this combination to probe mtCB1R and studied their energetic dynamics, demonstrating a potential link between specific neuronal functions and their energetic metabolism [12]. Immunogold labelling was also used by Katona and Freund to identify CB1R on other intracellular compartments such as Golgi and endoplasmic reticula [68].

STORM is another advance in optical imaging that is considered to be one of the most promising tools to image the ECS. For example, Dudok et al. employed STORM to study the synaptic ECS architecture and CB1R distribution with very high resolution, confirming its power to resolve cellular and subcellular molecule distribution at nanoscales [69]. Other interesting applications of this technique were shown by Zoldi and Katona, who were able to couple STORM imaging with behavioural pharmacology experiments, and Zhou et al., who investigated the receptor tyrosine kinase (RTK) transactivation mediated by CB1R [70,71].
ijms-24-15829-t003_Table 3Table 3Imaging tools used to study CB1R in the CNS, with particular focus on their respective technical advantages and disadvantages.TechniqueAdvantagesDisadvantagesReferencesRadioligands[^18^F] radioactive ligands coupled with PET ^1^ and autoradiographyNon-invasive in vivo or ex vivo toolTarget visualizationPharmacodynamic and pharmacokinetic dataNo need to validate antibodyAlternative to immunofluorescenceHigh tissue penetration and can be used for whole-body imagingRadiation exposureRelatively expensiveLow spatial resolution and long acquisition time[54,55,56][^11^C] radioactive ligands coupled with PET ^1^ and autoradiography Non-invasive in vivo or ex vivo toolTarget visualizationPharmacodynamic, pharmacokinetic dataNo need to validate antibodyAlternative to immunofluorescenceHigh tissue penetration and can be used for whole-body imagingFaster synthesis than [^18^F] probesRadiation exposureRelatively expensiveLow spatial resolution and long acquisition timeShorter half-life compared to [^18^F]Limits in quantifying the concentrations and kinetics[57,72]Optical microscopyFluorescence light microscopy (immunocytochemistryand immunohistochemistry)Can be used with low magnificationsLow costHigh sensitivityHigh resolutionHigh simplicityHigh speedCan be used to elucidate distributions from organ to subcellular levelLower resolution compared to other techniquesLimited tissue penetrationPotential interference of endogenous moleculesRequires antibody validation[61,62]Fluorescent probesReal-time monitoringSpatiotemporal precisionSimilar features of immunofluorescenceContinuous optimization of existing moleculesRequires a linker to separate the probe from the ligand[47]FRET ^2^Very high resolutionRelatively low cost compared to electron microscopyCan be used to study proteininteractionsLack of appropriate labelling for intracellular proteinsTarget molecules must be close to each other[48,63]Electron microscopy TEM ^3^Very high resolutionElement and structural compound data retrievalRelatively expensiveLaborious sample preparationHigh risk of contamination from sample preparation processesTime consuming[12,65,66,67,68]Super-resolution microscopySTORM ^4^High lateral, axial, and spatial resolution3D-STORMCan be used to study distributions at nanoscalesLarge data processingLow speedRequires specialized fluorophores[69,70,71]PET ^1^: positron emission tomography; FRET ^2^: fluorescence resonance energy transfer; TEM ^3^: transmission electron microscopy; STORM ^4^: Stochastic Optical Localization Microscopy.


### 2.3. CB2R

Radioligands can also be used to image CB2R; note the abundance of PET studies assessing its role in neuroinflammation (Table 4). To date, no CB2R radioligand has been approved due to poor specificity in the CNS, but some show significant potential [73]. Here, we review the main radioligand categories and cite the most promising candidates. CB2R radioligands are usually divided into five main subcategories based on their chemical class: oxoquinolines, oxadioles, thiophene and thiazole derivatives, pyridines, and indole derivates. Oxoquinoline compounds have significant potential for in vitro PET studies, but strong adverse effects impede their in vivo translatability. The most promising compound in this category is [^11^C] RS-020, as a hydrophilic derivative of [^18^F]RS-126. This radioligand showed significant stability in the brain of Huntington’s disease (HD) mouse models and human amyotrophic lateral sclerosis (ALS) spinal cord tissues [74]. Oxadiazole, pyridine, and indole derivatives show promising preliminary results in vitro but fail in vivo. In the class of thiophene and thiazole derivatives, the most promising radiotracer is [^11^C]A-836339 that has been exploited in Alzheimer’s disease (AD) mouse models to imagine neuroinflammation [75].

As for CB1R, multiple IHC and ICC techniques have been used to image CB2R coupled with light microscopy. The very first distribution studies with this tool detected CB2R expression in the spleen only; thus, it was classified as an exclusively peripheral protein. However, recent gene expression and immunocytochemistry studies have shown CB2R expression in the brain; thus, the exact distribution of CB2R remains a controversial topic. Wu and Wang used rabbit polyclonal antibodies to image the spatiotemporal changes of the receptor in the rat hippocampus, but an interesting specificity assessment by Zhang et al. demonstrated that available CB2R antibodies have poor specificity, bringing these and similar results into question [76,77].

Beyond canonical IHC and ICC, many different agonist or antagonist fluorophores have been used to study CB2R and have been gradually optimized to increase their efficacy. The fluorescent properties of biarylpyrazole derivatives have been exploited for this purpose. Bai et al. developed NIR-mbc94, which emits in the near-infrared (NIR) spectrum and can bind endogenous CB2R with high affinity in mouse-derived microglial and neuronal primary cultures [78]. An updated version was created by Wu et al., known as ZW760-mbc94, with significantly increased fluorescence intensity [79]. Other oxyquinoline and indole fluorophore derivates that show significant affinity and selectivity have mainly been used to assess inflammatory cells. Interestingly, derivatives of the THC molecular core have also been exploited for imaging studies, exemplified by Singh et al. who identified such a ligand and used it to successfully visualize CB2R in living cells with confocal microscopy [73,80].

FRET has also been used to assess whether CB2R can form heterodimers to understand its effects on downstream pathways. To that end, Mensching et al. investigated cAMP responses at the single cell level through CB2R [81].

Electron microscopy has also been employed to study CB2R. Onaivi et al. utilised immunoelectron microscopy to elucidate the CB2R subcellular location in rodent brains. Esteban et al. used a similar technique in AD mouse models, detecting the receptors on microglial cells [82,83]. Considering the controversy regarding CB2R expression in the brain, microscopy tools have mainly been exploited to study the receptor’s structure. Xing et al. described the 3D structure of CB2R-G_i_ and compared it to CB1R-G_i,_ showing overall similar interaction profiles between the two receptors. These structural results are critical to understanding CB2R function and, if complemented with further structural analyses and computational docking simulations, could help in synthetizing drugs to selectively target the receptor in CNS disorders [84]. To the best of our knowledge, super resolution techniques such as STORM have yet to be applied to CB2R [52].
ijms-24-15829-t004_Table 4Table 4Imaging tools used to study CB2R in the CNS, with particular focus on their respective technical advantages and disadvantages.TechniqueAdvantagesDisadvantagesReferencesRadioligands[^18^F] and [^11^C] radioactive ligands coupled with PET ^1^ and autoradiography Non-invasive in vivo or ex vivo toolTarget visualizationPharmacodynamic, pharmacokinetic dataNo need to validate antibodyAlternative to immunofluorescenceHigh tissue penetration and can be used for whole-body imagingRadiation exposureRelatively expensiveLow spatial resolution and long acquisition timeNo radioligand has been approvedSignificant adverse effects[74,75]Optical microscopyFluorescence light microscopy (immunocytochemistryand immunohistochemistry)Can be used with low magnificationsLow costHigh sensitivityHigh resolutionHigh simplicityHigh speedCan be used to elucidate distributions from organ to subcellular levelLower resolution compared to other techniquesLimited tissue penetrationPotential interference of endogenous moleculesRequires antibody validation[76,77]Fluorescent probesReal-time monitoringSpatiotemporal precisionSimilar features to immunofluorescenceContinuous optimization of existing moleculesRequires a linker to separate the probe from the ligand[78,79,80]FRET ^2^Very high resolutionRelatively low cost compared to electron microscopyCan be used to study proteininteractionsLack of appropriate labelling for intracellular proteinsTarget molecules must be close to each other[48,81]Electron microscopy TEM ^3^Very high resolutionElement and structural compound data retrievalRelatively expensiveLaborious sample preparationHigh risk of contamination from sample preparation processesTime consuming[82,83,84]PET ^1^: positron emission tomography; FRET ^2^: fluorescence resonance energy transfer; TEM ^3^: transmission electron microscopy.


### 2.4. Imaging the Endocannabinoids

Spatiotemporal distribution of eCBs remains elusive due to a lack of tools capable of directly visualizing them. This is due to the lipid nature of eCBs, the plethora of structural analogues in tissues, and the short half-life of functional eCBs [29,41]. To date, even the simplest immunofluorescence approaches are impeded due to a lack of commercial lipid-targeting antibodies.

Traditional metabolomics/lipidomics techniques such as liquid chromatography followed with mass spectrometry (MS) have been adapted to identify and quantify eCBs with high sensitivity and specificity but since these techniques often involve analyte extraction from the biological matrix, they offer very limited spatiotemporal resolution [85,86,87]. Nielsen et al. were able to exploit MS imaging (MSI) with matrix-assisted laser desorption/ionization (MALDI-MSI). This technique measures ions grid-wise across a 2D tissue matrix to detect high levels of brain MAG during the early phase of ischemia. However, they could not distinguish between 1-AG or 2-AG, and operated on the assumption that their analysis reflected the natural abundance of 2-AG over 1-AG in brain tissue [88] (Table 5).

Other methods have been used to indirectly assess spatiotemporal eCB dynamics such as electrophysiology or microdialysis coupled with pharmacological or genetic manipulations, but both present important limitations preventing precise measurement of the ECS in vivo [89]. Due to these critical limitations, even key assumptions regarding eCB signalling such as its duration (in the order of seconds) and spread (in the order of micrometres) have yet to be unambiguously confirmed [1].

Fortunately, new tools are continuously in development. High-specificity sensors developed through protein engineering have enabled, at least partially, bridging of these methodological gaps. Recently, Dong et al. developed such bioengineered fluorescent sensors based on GPCRs and circular-permutated fluorescent proteins (cpEGFP) to detect extracellular levels of dopamine, norepinephrine, acetylcholine, and serotonin [89]. Replicating this strategy, they created a new GPCR-activation-based eCB sensor known as GRAB_eCB2.0_ that is based on the human CB1R and cpEGFP. This sensor has high selectivity, sensitivity, and favourable kinetics for detecting AEA and 2-AG in both healthy and diseased neuronal cultures, brain slices, and specific brain structures such as the amygdala and the hippocampus. Furthermore, GRAB_eCB2.0_ can visualize eCBs’ dynamics during physiological and pathological processes in vitro and in vivo, opening a door to overcome current limitations in investigating eCB spatiotemporal dynamics.

### 2.5. Imaging eCB Metabolic Enzymes

Like CBRs, radiotracers are also frequently used to assess the ECS enzymes (Table 6). However, developing radioligands for MAGL has been very challenging due to low brain penetration and selectivity. One of the most promising radioligands to date, [^18^F]PF06795071, was used by Chen et al. and has a high binding specificity and a brain uptake consistent with MAGL distribution [90]. New strategies are continuously in development and may derive from the search for ligands with other functions. For example, He et al. created reversible MAGL inhibitors containing a morpholine-3-one scaffold as a potential treatment for neuroinflammation. Their design showed significant brain permeability and potential for neuroimaging applications [91].

FAAH radioligands are usually classified as either reversible or irreversible tracers based on their binding kinetics. Two of the most interesting examples to date are of an irreversible nature: [^11^C]CURB that has been utilised in a vast range of psychiatric, neurological disorders and addiction investigations [92], and [^18^F] FCHC that shows high biodistribution and brain penetration in rodents [93,94,95].

In contrast, DAGL and NAPE-PLD are preferentially imaged with fluorescent probes [96]. The Van Der Stelt group identified a selective NAPE-PLD inhibitor, LEI-401, which reduced the levels of the synthetic enzyme in a dose-dependent manner [97]. To assess DAGL, Rooden et al. discovered a fluorescent probe with good cell membrane permeability that was used to successfully image living cells [98].

Precise data regarding neuroanatomical localization and distribution of these enzymes are critical to elucidate the role of ECS signalling in the CNS. Light microscopy with immunocytochemistry is frequently employed to that end, as exemplified with studies that localized DAGLα to somatodendritic membranes and spines in the mouse brain [99,100]. FAAH distribution has been investigated with the same technique and was localized to proximal dendrites close to CB1R in the hippocampal pyramidal cells, cortex, and brainstem [101].

Immunogold TEM aided in precise localization of the main 2-AG metabolic enzymes. Gulyas et al. confirmed DAGLα localization to the postsynaptic dendritic spines and MAGL to presynaptic axon terminals, in accordance with their role in retrograde regulation of the synaptic activity [102]. Similarly, NAPE-PLD was detected mainly in neurites close to the synapse, while FAAH was found expressed in cytoplasmic vesicles, mitochondria, and endoplasmic reticulum [103].
ijms-24-15829-t006_Table 6Table 6Imaging tools used to study eCB metabolic enzymes in the CNS, with particular focus on their respective technical advantages and disadvantages.TechniqueAdvantagesDisadvantagesReferencesRadioligands[^18^F] and [^11^C] radioactive ligands coupled with PET ^1^ and autoradiography Non-invasive in vivo or ex vivo toolTarget visualizationPharmacodynamic, pharmacokinetic dataNo need to validate antibodyAlternative to immunofluorescenceHigh tissue penetration and can be used for whole-body imagingRadiation exposureRelatively expensiveLow spatial resolution and long acquisition time[90,91,92]Optical microscopyFluorescence light microscopy(immunocytochemistryand immunohistochemistry)Can be used with low magnificationsLow costHigh sensitivityHigh resolutionHigh simplicityHigh speedCan be used to elucidate distributions from organ to subcellular levelLower resolution compared to other techniquesRisk of limited tissue penetrationPotential interference of endogenous moleculesRequires antibody validation[99,100]Fluorescent probesReal-time monitoringSpatiotemporal precisionSimilar features to immunofluorescenceContinuous optimization of existing moleculesRequires a linker to separate the probe from the ligand[96,97,98]Electron microscopy TEM ^2^Very high resolutionElement and structural compound data retrievalRelatively expensiveLaborious sample preparationHigh risk of contamination from sample preparation processesTime consuming[102,103]PET ^1^: positron emission tomography; TEM ^2^: transmission electron microscopy.


## 3. The Best Genetic Models to Investigate the ECS

The functions and mechanisms of the ECS have been extensively investigated in physiological and pathophysiological conditions using transgenic and knockout (KO) animal models.

Varied vertebrates and invertebrates show differential patterns of CBR expression. The CB1R and CB2R orthologs that have been identified in specific animal models show high interspecies homology. This homology is particularly evident between humans and the most prevalent mammalian model organism, i.e., mice. We firstly present the available ECS genetic models in three of the most common laboratory model species: *Caenorhabditis elegans* (*C. elegans*), zebrafish, and mice. We cite advantages and disadvantages for each, towards equipping the reader with the needed information to choose the best model for their experimental conditions. As several comprehensive reviews for each separate model species already exist [104,105,106], this section aims to provide an update of the latest applications and discoveries in these models—with specific emphasis on CNS and CNS-disorder-related studies.

### 3.1. C. elegans as a Model

The nematode *C. elegans* is an important invertebrate model for a wide range of biological research applications due to its practical advantages over more complex species. Its small size and cost-effective maintenance make it practical for storage in limited laboratory spaces [107]. They produce multiple offspring in a 3-day reproduction cycle, while thousands of commercially available mutant strains facilitate genetic and pharmacological studies. Of particular importance to CNS research, *C. elegans* has a fully characterised nervous system which consists of 302 neurons with a completely sequenced genome [108].

#### 3.1.1. The Endocannabinoid System in *C. elegans*

Substantial research has focused on confirming the presence of a functional ECS in *C. elegans*. In 2008, it was first demonstrated that *C. elegans* possesses the ability to synthesize eCBs, specifically AEA and 2-AG [109]. These compounds were detected in the *C. elegans* Bristol N2, AB1 (Australian worms), CB4856 (Hawaiian worms), and TR403 strains (wild type). They were not observed in the *C. elegans* fat-3 mutant since they lack the necessary enzyme to produce the arachidonic acid precursor. Subsequently, orthologs of the enzymes involved in AEA biosynthesis and degradation were identified in *C. elegans*. Nape-1 (Y37E11AR.4) and nape-2 (Y37E11AR.3) were identified as two orthologues of mammalian NAPE-PLD, capable of generating NAEs in vitro. A BLAST analysis also identified an ortholog of FAAH in *C. elegans* [110,111,112]. While orthologs of enzymes involved in the synthesis of 2-AG have been proposed, such as DAGLα and DAGLβ, they have not been confirmed to date. Putative CBRs for AEA and 2-AG have also been identified in *C. elegans* [113]. Sequence alignments were performed in the early 2000s to identify homologous sequences for mammalian CB1R and CB2R. Results revealed substitutions at critical amino acid residues and suggested that none of the CBR candidates in *C. elegans* could be considered homologous to those in humans [114]. However, new approaches using phylogenetic trees have proposed the nematode sequence C02H7.2 as a candidate for CB1R [110] (Figure 3). These discrepancies may arise from differences in the genetic tools used and underscore the need for further investigations in this species.

Despite the lack of a definitive CBR, there is evidence that the ECS contributes to several processes in *C. elegans*, such as reproductive development, cholesterol transport, axon regeneration, and behaviour [115]. The molecules AEA and EPEA have been shown to inhibit axon regeneration in a manner dependent on the AEA receptor candidates, NPR-19 and NPR-32 [113,116]. These findings suggest the presence of putative receptors that could modulate some eCB-dependent outcomes. NPR-19 is of special interest since both 2-AG and AEA are shown to bind to this receptor. More recently, eCBs were shown to differentially modulate serotoninergic, dopaminergic, and cholinergic systems, evidencing the potential of this worm model to study cannabinoid signalling and its possible correlated changes in behaviour [117,118,119].

#### 3.1.2. Genetic Approaches

*C. elegans* is a popular model in biomedical research due to the relative ease of manipulating its gene expression, coupled with the advantages previously described [120]. Despite the presence of typical eCBs and the orthologous genes for their metabolism, the controversy surrounding existing data may explain a lack of genetic models for the ECS in *C. elegans*. However, these discrepancies should be reconsidered since recent data demonstrate the functionality of ECS in this worm model. Many of the practical advantages of these models are also shared with zebrafish, whose genome is well characterised and has many orthologs in the human genome. Thus, the zebrafish is widely considered to be a reasonable model organism for ECS research.

### 3.2. Zebrafish as a Model for ECS Studies

*Danio rerio*, commonly known as zebrafish, are small tropical fish native to Southeast Asia. Despite the limitations of using primitive fish as a model for complex human diseases, recent genetic advances underscore the numerous advantages that have established zebrafish as an increasingly important model. Some of these advantages include its rapid development and fecundity, well-characterised genome and orthologs to human genes, ease of genome editing, scalability for high-throughput pharmacological screens, as well as cost- and space-efficiency [121,122].

#### 3.2.1. The Endocannabinoid System in Zebrafish

Phylogenetic studies have revealed a high degree of similarity between the ECS in zebrafish and that of humans [110,123] (Figure 3). In contrast to other invertebrates, the ECS of zebrafish possesses orthologs of almost all human ECS components (with notable exceptions such as NAAA), along with similar expression patterns [106,124,125]. The degree of identity and coverage between zebrafish and humans has been investigated by Klee et al., revealing the most conserved genes to be those encoding CB1R and DAGLα, with 72% and 67% similarity, respectively [123]. It is also believed that these animals produce similar endogenous CBR ligands to mammals. Typical CBR ligands such as THC, HU-210, WIN55212-2, CP55940, and AEA interact with CB1R in the zebrafish brain, stimulating G-protein activity as expected [126]. Consequently, the high degree of ECS homology between zebrafish and humans, coupled with its unique advantages, positions it as a promising genetic model.

#### 3.2.2. Genetic Approaches for ECS Manipulation in Zebrafish

The first zebrafish model for ECS genetic alteration was developed 15 years ago through morpholino-based knockdown of *cnr1* [127]. Morpholino phosphonodiamidite antisense oligomers (morpholinos) are stable nucleic acid analogues that have been extensively used in gene knockdown studies. They contain DNA bases designed to bind complementary RNA sequences in the target region and prevent the binding of other large molecules. Their functionality depends on the biological function of the specific RNA sequence they target, blocking association of either ribosomal or critical mRNA processing proteins that lead to inhibition of gene expression [128].

Thanks to recent breakthroughs in genomic technology, the zebrafish genome has been well characterised and extensively modified. Thus, more ECS models were created using cutting-edge gene-editing technologies such as transcription-activator-like effector nucleases (TALENs) and clustered regularly interspersed short palindromic repeat/Cas9 (CRISPR/Cas9) [129,130]. These techniques have enabled precise gene targeting at various developmental stages, from juveniles (embryos and larvae) to adults. Additionally, some researchers have also employed the Tet-off transgenic system to achieve inducible gene expression, offering temporal control over gene expression [131]. This section of the review will discuss the major discoveries facilitated with these techniques and the methods employed to create these genetic models. The table below presents a general overview of pioneering studies involving zebrafish genetic models in ECS research (Table 7).

#### 3.2.3. CB1R Models in Zebrafish

Most zebrafish models related to the ECS focus on knockouts of the *cnr1* gene. Table 8 displays the targeting sequences for the *cnr1* gene across various studies.

The first CB1R-KO model was created by Watson et al. using morpholinos. Watson et al. used two different morpholinos, both of which blocked CB1R translation [127]. These studies revealed the impact of ECS perturbations on early neuronal development, as CB1R knockdown resulted in defects in axonal growth and fasciculation. Over time, morpholinos targeting different sequences within the CB1R gene have been developed, all of which target positions ranging from 907,700 to 909,000 kb on chromosome 20 in zebrafish that encode exon 2 of *cnr1* [140].

Nishio et al. investigated the role of ECS in fasting, as CB1R has been proposed as a pathway through which endocannabinoids serve as orexigenic signals [132]. While the literature suggests that the CB1R blockade leads to reduced food consumption in animals [141,142], the underlying mechanisms are still incompletely understood. Nishio et al. studied how CART (cocaine- and amphetamine-related transcript) might participate in the hypothalamic mediation of orexigenic endocannabinoid action. CART is a neuropeptide related to reward and feeding that produces similar effects to the psychostimulants for which it was named [143]. Utilizing two different antisense morpholinos targeting different regions of CB1R mRNA, they showed that CB1R loss decreased *cart3* and *cart2a* expression, indicating that CB1R acts upstream of CART and influences appetite through its downregulation [132]. CB1R activation in fasted animals also leads to a decrease in CART expression. Taken together, these findings suggest an intricate connection between the ECS, CART, and appetite regulation that is extremely context-specific. Moreover, these data improve the understanding on regulation of food intake via CB1R and can help to reduce side effects associated with the use of CB1R antagonists [144]. Shimada et al. investigated the role of the CB1R in appetite regulation, demonstrating that *cnr1* knockdown in zebrafish decreased spontaneous locomotion and suppressed appetite, observations resembling those seen in CB1R-inhibited mammals [133].

Finally, Liu et al. showed that *cnr1* knockdown resulted in a smaller liver with fewer hepatocytes and decreased liver-specific gene expression and cell proliferation. They used two different approaches to create *cnr1^−/−^* mutants: a morpholino-based approach and TALENs [145].

While most models involve *cnr1* knockdown, Pai et al. employed the tetracycline-inducible expression system to overexpress CB1R in the liver. This system allows for relatively precise, reversible, spatially restricted, and quantitative regulation of target gene expression in transgenic animals [146]. The results showed lipid accumulation in CB1R transgenic zebrafish without doxycycline treatment (expressing high levels of CB1R) and the loss of lipid accumulation in zebrafish treated with doxycycline (lacking CB1R protein).

Finally, Fin et al. utilised the CRISPR/Cas9 technique to study the function of the *cnrip1* gene, generating a line with predicted null mutations in *cnrip1a* and *cnrip1b* genes [137]. Both genes are expressed primarily in the brain and spinal cord, and they were believed to interfere with CB1R function. However, both mutants lacked any noticeable phenotype.
ijms-24-15829-t008_Table 8Table 8Targeting sequences for c*nr1* used in zebrafish CB1R-knockdown studies.Target Sequence of *cnr1*
AnnotationsReferences5′-CGGACTTTGAGGCCGGGAACAGCAT-3′Translation-blocking[127]5′-CTAGAGGAAACCTGTCGGAGGAAAT-3′Translation-blocking[127,139]5′-GAATGACTACGCTTACATGGACATC-3′Target the 5′UTR[132]5′-AACAGCATGGTCAGAGATGCTCTAG-3′Translation-blocking[132]5′-GTGCTATCAACAACATACCTTTGTG-3Splice-blocking[133,145]5′-CTTTGAGGCCGGGAACAGCATGGTC-3Splice-blocking[145]5′-GAACAGCATGGTCAGAGATGCTCTA-3′Translation-blocking[136]5′-TCAGAACCATCACCTCCG-3′5′-TCAGAACCATCACCTCCG-3′Target first exon[146]


#### 3.2.4. CB2R Models in Zebrafish

Several zebrafish *cnr2*-knockout models have also been developed, as indicated in Table 9, which presents the gene targeting sequences across these studies. Most of the selected studies (Table 7) employed morpholino injections to generate these mutants. In contrast to *cnr1-*directed morpholinos, morpholinos against *cnr2* target two distinct regions that correspond to the two different exons of the gene: from 33,111,900 to 33,112,100 kb and from 33,113,900 to 33,114,300 kb on chromosome 16 [147]. While no noticeable differences due to this differential targeting were observed in the cited studies, it is noteworthy to keep in mind while considering any heterogeneity in results that may arise between these two strategies.

Nishio et al. also utilised a morpholino targeting the *cnr2* AUG region, showing that its blockade did not abolish the rimonabant-induced increase in yolk size, suggesting limited CB2R involvement in feeding behaviour [132]^.^

Previous research on the ECS in humans and mice has demonstrated that CB2R regulates immunity through various mechanisms, including modulation of distinct leukocytes [148,149]. To study the role in zebrafish leukocyte migration, a context-dependent assembly zinc-finger nuclease (CoDA ZFN) technique was used to induce CB2R knockdown. This approach also enabled tracking the migration of leukocytes, as monocytes/macrophages and neutrophils expressed a fluorescent protein (EGFP) [134]. The study revealed that CB2R inhibition enhanced leukocyte migration.

eCBs can modulate the function of adult hematopoietic stem and progenitor cells (HSPCs) [150], but their effect on embryonic HSPCs remains unknown. Esain et al. showed that morpholino-based *cnr2* knockdown in juvenile zebrafish reduced HSC count and *runx1* expression, a transcription factor necessary for the HSPC development. These results provide evidence of CB2R’s critical role in regulating HSPCs and reveal a novel role for eCBs during vertebrate haematopoiesis [135]. In another study, loss of CB2R led to altered biliary anatomy and lipid processing and increased susceptibility to hepatic steatosis [134].

Notably, the CRISPR/Cas9 gene editing system has also been used to generate CB2R-KO zebrafish strains. Acevedo-Canabal et al. generated one such strain to show that zebrafish lacking CB2R exhibit distinct anxiety-like behaviours [138]. While CB2R expression in the brain is still a contentious topic, a previous study in mice indicated a potential role of CB2R activation in anxiety-like behaviours. It is worth noting that drugs were administered systemically and a possible peripheral effect was not ruled out [151]. Nevertheless, the mechanisms underlying this observation in both animal models still require extensive investigation.
ijms-24-15829-t009_Table 9Table 9Targeting sequences of the *cnr2* gene used in zebrafish CB2R-knockdown studies.Target Sequence of *cnr2*AnnotationsReferences5′-CTGCTCTTGTGTGGTCAAAACCATG-3′CB2R AUG[132]5′-ATGGCGTTTACGGGCTCTGT-3′5′ end of exon 2[138]5′-GCCATGAAACAAACAGTACCTGTGG-3′Splice-blocking[135,145]5′-GTTCCAGTTTGTTCTCCATTTTCCC-3′Translation-blocking[145]


### 3.3. Mouse Models

Mouse models are undoubtedly the most prevalent strategy for studying the individual ECS components, with several previous reviews summarizing their contribution to our understanding of the system [107,152,153]. As the standard laboratory animal model with the highest ECS homology to humans and a comparatively complex CNS and behavioural paradigm, they are indispensable to preclinical endocannabinoid research and ECS-related drug development for CNS disorders. Here, we present an updated summary of the available transgenic mouse models for each ECS component, with a focus on their contribution to the literature on the system’s role in the CNS. Note that the abbreviations used to designate the different models are purely for ease of reference in this review and not necessarily the official strain designations.

#### 3.3.1. CB1R Mouse Models

Several CB1R-knockout mouse models have been generated with different genetic contexts (Table 10). The first reported CB1R-knockout (CB1R-KO) model was published by Ledent et al. [154]. In this model, embryonic stem (ES) cells from a 129-mouse strain were edited by replacing the first 233 codons of *Cnr1* with a phosphoglycerokinase promoter driving an aminoglycoside phosphotransferase expression (PKG-neo) cassette, which lead to excision via homologous recombination. Subsequently, these mice lacking CB1R were outbred for several generations on a CD1 background. The number of backcrossed generations varies between studies, ranging from 5 to 30 [152,153,155,156,157].

In the same year, another CB1KO model was introduced by Zimmer’s group [142,158]. They utilised a similar PKG-neo recombination strategy, only replacing the coding region of the *Cnr1* gene between amino acids 32 and 448. These chimeric mice were subsequently bred with C57BL/6J animals for more than 10 generations, resulting in mutants with a congenic C57BL/6J background [159,160,161,162]. Marsicano et al. later generated a new CB1R-null mutant line using the Cre-lox system for site-directed recombination [163]. Mice bearing a floxed-neo *Cnr1* allele were crossed with transgenic mice expressing Cre recombinase ubiquitously, leading to its excision. Subsequently, mice were bred for five generations into a C57BL/6N background [164,165,166].

Robbe et al. published another method to generate CB1R-KO mice [167]. Briefly, they replaced part of the upstream noncoding region down to the sixth transmembrane region of the receptor with a neomycin resistance cassette, leading to a non-functional receptor. At the end, CB1R mutant mice were backcrossed to C57BL/6 mice.

Finally, another available mutant line was generated by Ibrahim et al. in collaboration with Deltagen [168]. They replaced the CB1R coding sequence encompassing bp 26-1249 with the IRES-LacZ-Neo-pA cassette on a 129/SvJ genetic background [168,169].

In summary, at least five different mutant lines have contributed to the study of the role of CB1R in the CNS, although others are available that have not been widely used for CNS studies [104]. Despite differences in mutation techniques and genetic backgrounds, these investigations have yielded similar and congruent results (Table 10). Overall, these mutant mice exhibited increased mortality and more severe seizures [142,164], anxiety-like behaviours [153,155], reduced locomotor activity [142,162], accelerated cognitive decline [159,160,161,162], and impaired memory extinction processes [163,165,166]. Interestingly, CB1R was proven to be dispensable for memory extinction in appetite-motivated learning tasks, whereas it presumably plays a crucial role in fear extinction, primarily via habituation-like processes [166].
ijms-24-15829-t010_Table 10Table 10Global CB1R genetic mouse models.Genetic BackgroundGeneral PhenotypeOutcomeReferencesSpontaneous Behaviour CD1(Ledent group)Significantly higher anxiety-like and aggressive behaviour, increased affected maternal care Increased locomotor activity and time spent exploring unknown objects; decreased spontaneous alteration in the Y-maze[154]Increased anxiety-like behaviour[170]Increased AMT ^1^ and FAAH ^2^ activity with age; mild-anxiety-like behaviour of young mice compared to old mice[152]Increased anxiety-like effect under unfamiliar environment [171]Increased anxiety-like behaviour in the elevated plus maze[155]High level of anxiety with different types of anxiogenic stimuli under unfamiliar conditions[153]No significant alterations in anxiety-like behaviours under total darkness conditions [172]Delayed pup retrieval and fewer ultrasonic vocalizations[173]Higher levels of offensive aggression when housed in group[174]C57BL/6J(Zimmer group)Hypoactive, increased mortality, reduced anxiety Reduced locomotion and rearing in open-field test[158]Increased mortality rate and ring catalepsy; reduced locomotor activity and hypoalgesia[142]Less burying behaviour and fewer contacts with the probe in the shock-probe burying test [175]C57BL/6N(Lutz group)Aversiveness-dependent anxiogenic-like phenotype and acute fear response, enhanced contextual fear memory, increased wakefulnessReduced spontaneous caloric intake and decreased body weight[176] (p. 200)Sustained fear response only after application of an intense foot shock (0.7 mA and 1.5 mA)[166]Lack of within-session extinction during permanent tones and repeated tone presentations at variable intervals[177]Decreased distance covered in the active and in the inactive phases of the cycle on wheel running activity[178]Disrupted classical fear conditioning pattern by favouring passive responses [179]Increased fear expression abolished by repeated social stress[180]Enhanced freezing levels in the conditioning context and increased contextual fear after high-intensity conditioning foot shock (1.5 mA)[181]Increased cortical excitability and reduced NREM ^3^ sleep and NREM ^3^ bout duration[182]More time awake and less time in NREM ^3^ and REM ^4^; slower EEG ^5^ theta rhythm during REM ^4^ and habituated more rapidly to the arousing effect of the cage-switch test[183]Hypoactivity, impaired eyeblink, and normal cerebellum-dependent locomotor coordination and learning[162]In vivo response to drugsCD1(Ledent group)Insensitive to THC and CBD, sensitive to nicotine, ethanol, cocaine, and amphetamineInsensitive to THC-induced antinociceptive properties, reduced horizontal activity, and decreased rectal temperature [154]Sensitive to SR141716A anxiety reduction [155]Enhanced nicotine-induced antinociceptive effects and absent rewarding effects[184]Decreased ethanol self-administration with increased sensitivity to its acute intoxicating effects[185]Decreased ethanol-induced conditional place preference and increased striatal dopamine D2 receptors[186]Reduced ethanol self-administration and ethanol-conditioned place preference[187]Decreased locomotor responses to cocaine and D-amphetamine[188]Insensitive to CBD-induced anxiolytic actions[157]C57BL/6J(Zimmer group)Insensitive to cannabinoid drugs and sensitive to cocaine and ethanol; absence of ethanol withdrawal Insensitive to THC-induced ring catalepsy, hypomobility, and hypothermia[142]Insensitive to THC-, WIN 55,212-2-, and methanandamide; disruption in the working memory task[159]No enhancement of growth rates or survival after CP55,940, WIN55,212-2, or 2-AG administration[189]Reduced voluntary alcohol consumption and absent alcohol–dopamine release in the nucleus accumbens[190]Absence of ethanol withdrawal symptoms and of foot-shock stress-induced alcohol preference[191]Reduced ethanol preference and insensitive to SR141716A-induced reduction in ethanol preference in young mice[192]Insensitive to THC- and O-1812-induced decrease in lever press; sensitive to methanandamide-, ethanol-, and morphine-induced decrease in lever press[193]Abolished CP55,940-induced antinociceptive effects and associated motor deficits[194]Absent THC-induced expression of ΔFosB in the striatum[195]Sensitive to the locomotor-stimulant effects of RTI-371 (a cocaine analogue)[196]Absence of ethanol-induced activation of caspase 3 and of reduction in DNMT1 ^6^, DNMT3A ^6^, and DNA methylation[197]C57BL/6N(Lutz group)Sensitive to KA ^7^ and insensitive to THC, reduced sensitivity to rewarding properties.KA ^7^ injection induced more severe seizures and decreased survival rate[164]Decreased sucrose consumption under operant conditions or a two-bottle free choice; decreased sensitivity to rewarding properties of sucrose[198]Insensitive to cannabinoid-induced neurosphere generation[199]Insensitive to THC-induced tetrad effects[200]Insensitive to THC-induced increase in pregnenolone in the nucleus accumbens[201]Learning and aging CD1(Ledent group)Defective neurogenesis, increased aggressiveness and conditioned responsesIncreased aggressive response, higher sensitivity to exhibit depressive-like responses, and increased conditioned responses in the active avoidance model[202]Enhanced retention of the habituation task[203]Reduction in bromodeoxyuridine-labelled cells in dentate gyrus and subventricular zone[204]C57BL/6J(Zimmer group)Impaired extinction, age-related memory decline, accelerated decline in cognitive functionIncreased perseverance in a reversal task [159]Similar or better performance on 6–7-week-old mice and worst performance on 3–5-month-old mice in several learning and memory paradigms [205]Impaired extinction process in the Morris water maze using a spaced extinction procedure[206]Enhanced habituation (non-associative learning) displayed with decreased number of ambulations [207]Impaired delay eyeblink conditioning performance[208]Deficits in a sensory-selective reinforcer devaluation task[209]Improved performance in the Morris water maze at 6 weeks old and inferior performance at 12 months old[161]Superior learning ability in the eight-arm radial maze at 2 months old and impaired performance at 12 months old[160]C57BL/6N(Lutz group)Impaired extinction of aversive memories and of fear, impaired habituation Strongly impaired short-term and long-term extinction in auditory fear-conditioning tests, with unaffected memory acquisition and consolidation[163]Normal extinction of the stimulus–response association in an appetitively motivated learning task[165]Severely impaired in extinction and in habituation of the fear response to a tone[166]AMT ^1^: membrane transporter; FAAH ^2^: fatty acid amide hydrolase; NREM ^3^: non-rapid eye movement; REM ^4^: rapid eye movement; EEG ^5^: electroencephalogram; DNMT ^6^: DNA methyltransferases; KA ^7^: kainic acid.


#### 3.3.2. Conditional CB1R Deletions

CB1Rs are broadly expressed in the CNS with varied distribution in different neuronal and glial subpopulations [210,211,212]. To elucidate the specific role of CB1R in these subpopulations, numerous mouse lines with cell-type-specific deletions have been generated (Table 11). However, one must consider that the genetic approaches used to create these mutant lines may introduce potential confounding factors. For instance, in Cre-induced recombination approaches, the use of specific regulatory sequences to activate Cre recombinase expression may produce recombination patterns different from those predicted.

A mouse line containing the cnr1-floxed-neo construct on a C57BL/6N background generated by Marsicano et al. has been widely utilised to produce conditional CB1R-KO (cKO) in different cell types. It was created by introducing two loxP sites flanking the *Cnr1* coding exon, before removing the FRT-PKG-Neo selection cassette by crossing these animals with Flipase-deleter mice. Subsequently, this line was crossed with mouse strains expressing Cre recombinase under the control of cell-type-specific promotors with different regulatory sequences [164].

In the CB1R^CaMKIIαCre^ line, CB1Rs are deleted in all principal neurons of the forebrain [164], which has provided insights into the ECS’s roles in protecting against acute excitotoxicity and acute fear adaptation [164,213,214]. A transgenic mouse line lacking CB1R in GABAergic neurons, the Dlx5/6-Cre line, was generated [213]. To achieve specific Cre-recombinase expression resembling Dlx5/Dlx6 gene patterns, researchers utilised intergenic enhancer sequences l56i and l56ii from zebrafish dlx5a/dlx6a. Monory et al. also generated a mouse line lacking CB1 receptors in glutamatergic cortical neurons: the NEX-Cre line [213].

Several other conditional mouse models relevant to the CNS have been generated. In these strains, the Cre-dependent CB1R line was crossed with various Cre expressing lines, generating the following CB1R-cKO lines: (1) D1-CB1KO, with dopamine receptor D1A gene (*Drd1a*)-driven Cre expression; (2) Gabra6-CB1KO, with GABAA-receptor-α6-driven Cre expression (limited to cerebellar granule cells); (3) GFAP-CreERT2, with GFAP (gfap)-driven Cre expression; (4) Sim1-CB1KO, in which Cre recombinase is controlled by the single-minded 1 (*Sim1*) transcription factor; (5) sns-CB1KO, with the transcriptional factor SNS (Nav1.8) gene driving Cre expression; and (6) TPH2-CreERT2-CB1KO, in which the Cre recombinase line was expressed under the regulatory sequences of the mouse tryptophan hydroxylase 2 (*Tph2*) gene locus [36,180,200,215,216].

The main outcomes of the above-mentioned CB1R-cKO mouse lines are summarized in Table 11.
ijms-24-15829-t011_Table 11Table 11Conditional CB1R genetic mouse models.Strain DesignationCell-Type-Specific DeletionOutcomeReferencesCB1R^CaMKIIa-Cre^Forebrain principal neuronsMore severe KA ^1^-induced seizures and decreased survival[36,164] (p. 200)Sustained fear response only after intense electric shock [214]CB1R^Dlx5/6-Cre^GABAergic neuronsNo change in KA ^1^-induced seizures [213]Impaired target selection of cortical GABAergic interneurons[217]CB1R is localized on presynaptic boutons of about 30% in alBNST ^2^[218]Neuronal loss and increased neuroinflammation in the hippocampus [161]Abolished anxiogenic effect under a high-dose treatment of CB1R agonist (CP-55940)[219]Conserved impairment of SWM ^3^ and in vivo LTD ^4^ of synaptic strength at CA3-CA1 synapses caused by an acute exposure to exogenous cannabinoids[36]Abolished CB1R agonist (CP-55940)-induced increase in HVS ^5^[220]Insensitive to quinolinic-acid-induced neurotoxicity[136]Insensitive to THC-induced memory impairment in novel object recognition[221,222]CB1R^Nex-Cre^Glutamatergic cortical neuronsAberrant fasciculation and pathfinding in both corticothalamic and thalamocortical axons[223]Absence of neuronal loss and increased neuroinflammation in the hippocampus[161]Unbalanced neurogenic fate determination[224]Conserved impairment of SWM ^3^ and in vivo LTD ^4^ of synaptic strength at CA3-CA1 synapses caused by an acute exposure to exogenous cannabinoids[36]Reduced decrease in fast ECoG ^6^ oscillations and stronger cannabinoid-induced increase in HVS ^5^[220]Sensitive to excitotoxic damage induced with quinolinic acid administration [136]CB1R^D1-Cre^Neurons expressing dopamine D1 receptorsInsensitive to THC-induced catalepsy[200]Abolished CB1R agonist (CP-55940)-induced increase in HVS ^5^[220]CB1R^sns-Cre^Dorsal root ganglia neuronsReduced LTD ^4^ at dorsal horn nociceptor synapses[215]CB1R^Gabra6-Cre^Cerebellar granule cellsAbolition of short-term plasticity at parallel fibre synapses and lack of LTD[225]Activated cerebellar microglia and increased cerebellar neuroinflammation[226]Normal eyeblink conditioning and normal cerebellum-dependent locomotor coordination and learning[162]CB1R^Gfap-CreERT2^AstrocytesAbolished impairment of SWM ^3^ and in vivo LTD ^4^ of synaptic strength at CA3-CA1 synapses[36]Impaired object recognition memory and decreased LTP ^7^ at CA3-CA1 synapses[37]CB1R^Sim1-Cre^Neurons expressing Sim 1 ^8^ (hypothalamus and mediobasal amygdala)Increased locomotor activity in open field, unconditioned anxiety, and cued fear expression under basal conditions[216]CB1R^Tph2-CreERT2^Central serotoninergic neuronsAnxiety and decreased cued fear expression[180]KA ^1:^ kainic acid; alBNST ^2^: anterolateral bed nucleus of the stria terminals; SWM ^3^: spatial working memory; LTD ^4^: long-term depression; HVS ^5^: thalamocortical high-voltage spindles; ECoG ^6^: neocortical electrocortigrams; LTP ^7^: long-term potentiation; Sim1 ^8^: single-minded 1.


#### 3.3.3. MtCB1R Models

Until 2016, studies of mtCB1R disruption were conducted in global CB1R-KO models or conditional knockouts for GABA- and glutamatergic neurons [12,24]. To date, only a few more specific models have been generated to assess mtCB1R (Table 12).

Hebert-Chatelain et al. showed that a specific mutant version of CB1R protein lacking the first 22 amino acids (DN_22_-CB_1_) maintains plasma-membrane-associated but not mitochondrial CB1R function [24]. Soria-Gomez et al. replaced the wild-type (WT) *Cnr1* gene with the coding sequence of DN_22_-CB_1_, generating a knock-in mutant mouse line (DN_22_-CB1R-KI). Using immunogold-labelling electron microscopy, [35S] GTPγ binding assays, and respirometry, they demonstrated that the model specifically impairs cannabinoids’ effects on the mitochondria, leaving all classical cell surface CB1R-related parameters unchanged or reduced [27]. Han et al. generated a tamoxifen-inducible model lacking CB1R in astrocytes. To this end, they crossed *Cnr1*-floxed mice with transgenic ones expressing Cre-ERT2 under the control of the human GFAP promoter, generating the GFAP-CB1R-KO strain. This strain showed a 79% reduction in CA_1_ astrocytes labelled with a CB1R antibody compared to WT counterparts [36]. Using this strain, Jimenez-Blasco et al. showed a link between astroglial mitochondria and cannabinoid effects [25].
ijms-24-15829-t012_Table 12Table 12mtCB1R mouse genetic models.Genetic BackgroundGeneral PhenotypeOutcomeReferencesC57BL/6NDN_22_-CB1R ^1^-KISpecific impairment of mtCB1R and of cannabinoid effects on mitochondrial dynamics but no influence on CB1R general functions mtCB1R decreases synaptic activity and induces catalepsy[27]mtCB1R mediates corticosterone-induced memory impairment[26]Astroglial mtCB1R reduces mitochondrial respiration with complex I destabilization[25]C57BL/6NGFAP ^2^-CB1R-KO Specific impairment of mtCB1R and of cannabinoid effects on mitochondrial dynamics but no influence on CB1R general functions Astroglial mtCB1R reduces mitochondrial respiration with complex I destabilization[25]DN_22_-CB1R ^1^: mutant version of CB1R protein lacking the first 22 amino acids; GFAP ^2^: glial fibrillary acidic protein.


#### 3.3.4. CB2R Mouse Models

As previously stated, CB2R is considered to be primarily a peripheral protein. It has mainly been detected in cells of an hematopoietic origin in the spleen, thymus, lymphocytes, and macrophages, while its expression in the brain is still under debate.

To date, there are two available mouse models for global CB2R-KO, both on a C57BL/6J background. Buckley et al. created a model (referred to here as CB2R^−/−Buk^) in which 341 bp of the coding exon were replaced with a neo selection cassette. Deltagen developed the other strain (referred to here as CB2R^−/−Del^) by deleting 334 bp via homologue recombination with a “Neo555T” construct, which was also repeated on a C57BL/6N background [104,227]. Both models have been extensively used to study the effects of CB2R depletion in neurodegeneration, pain, behaviour, addiction, as well as other peripheral contexts such as skin disorders, the immune and cardiovascular systems, and the liver [104] (Table 13).

While the majority of CB2R-KO studies to date pertain to one of the two models above, conditional transgenic models have also recently been utilised with reasonable success. A mouse line depleting CB2R under the control of a prion promoter was generated by Garcia-Gutierrez and Manzanares [228], which was used to show that CB2R depletion reduces hyperalgesia in neuropathies as well as anxiety-like behaviours [229].

Recently, Lopez et al. generated a CB2R reporter line (CB2R^−/− Lop^) with EGFP expressed under the control of the *Cnr2* gene promoter through the insertion of an internal ribosomal entry site. Furthermore, the entire *Cnr2* exon was flanked by loxP sites, facilitating conditional knockout creation through validated Cre-expressing models in future studies [230].
ijms-24-15829-t013_Table 13Table 13CB2R genetic mouse models.Genetic BackgroundGeneral PhenotypeOutcomeReferencesNeurodegeneration, neuroinflammation, and synaptic plasticityC57BL/6JCB2R ^−/− Buk^Increased neurodegenerative symptoms, impaired neuroprogenitor proliferation, impairment of neuroprotective proteinsProtective role for CB2R in experimental autoimmune encephalitis[231]Augmented multiple sclerosis severity (similar to pharmacological inhibition)[232]Link between CB2R and the onset of Huntington’s disease[233]Amelioration of Alzheimer’s disease-like pathology[234]CB2R-mediated modulation of cocaine action[235]Incomplete activation of microglia in neuroinflammation[236]C57BL/6JCB2R ^−/− Del^Higher corticosterone levels after stress in the prefrontal cortex (PFC), higher hippocampal and PFC neuron excitability CB2R activation mediates PFC neuron excitability[237]Chronic CB2R activation in the hippocampus increases excitatory synaptic transmission[238]C57BL/6JCB2R^−/− Lop^Increased neurodegenerative symptoms,agonist treatment can reduce inflammatory phenotypesDevelopment of a new reporter mouse line and involvement in neuroinflammation[230]Tau protein levels increase CB2R during early stages of neurodegeneration[239]CB2R depletion reduces inflammatory pain behaviours and markers of neuroinflammation[240]**Nociception and neuropathic pain**C57BL/6JCB2R ^−/− Buk^Normal cannabidiol analgesia, altered opioid receptor expressionNeuropathic pain is mediated by glycinergic neurons[241]CB2R mediates analgesic effects in neuroinflammation, neuropathies[242]C57BL/6JCB2R ^−/− Del^Reduced morphine analgesia, no effect of AM1710 on paclitaxel-induced allodyniaChronic CB2R activation reverses paclitaxel-induced neuropathy[243]Activation of CB2R alone or with CB1R decreases neuropathic-pain-related behaviour in mice[244]Possible mechanism to suppress chemotherapy-induced neuropathy[245]**Behaviour**C57BL/6JCB2R ^−/− Buk^Impaired memory consolidation, schizophrenic behavioural phenotypesInduces schizophrenia-related behaviours such as locomotor activities, anxiety- and depressive-like behaviours, and cognitive deficits[246]CB2R role in cognitive processes, particularly in short- and long-term memory[247]CB2R is expressed in red nucleus glutamate receptors and modulates motor behaviour[248]C57BL/6JCB2R ^−/− Del^Impairment of contextual long-term memory, enhancement of spatial working memory, decrease in neuropathic pain-related behaviour in miceCB2R plays different roles in regulating memory with different outcomes depending on the brain areas[249]CB2R^−/− Buk^: model created by Buckley et al.; CB2R^−/− Del^: model created by Deltagen; CB2R^−/− Lop^: model created by Lopez et al.


#### 3.3.5. CB1R × B2R Genetic Models

To tease apart the individual and combined effects of CB1R or CB2R, individual KO lines have been crossed to generate double KO models. However, few studies with such models focusing on the CNS have been reported to date. Of the main phenotypic features described for the double KO in relation to CNS function include reduced 2-AG-mediated locomotor activity and loss of THC-mediated analgesia [104,250]. Zador et al. utilised the double KO model to study the CBRs’ influence on the forebrain μ-opioid receptor, demonstrating that they work independently [49]. Recently, Ward et al. assessed the effects of the double KO on ischemia, demonstrating that depletion of both receptors did not exacerbate the ischemic process, contrarily leading to an improvement in the animals’ phenotype [251].

#### 3.3.6. Genetic Mouse Models of eCB Metabolic Enzymes

Endocannabinoid metabolism is characterised by functional redundancy, given that several enzymes and precursors can contribute to their synthesis and degradation [29,35,252,253]. Furthermore, the involved components participate in multiple aspects of phospholipid metabolism (functional promiscuity). Despite the resulting complications, several genetic models for studying the individual eCB metabolic proteins have been employed to date.

2-AG is a stronger CB_1_R agonist than AEA and also more abundant; therefore, most studies targeting the ECS have focused on modulating 2-AG tone [41,254]. Nevertheless, their actions are thought to be summative towards CBR agonism, with significant overlap in metabolic profiles produced by inhibiting 2-AG or AEA breakdown [85,255]. While the foundational models for genetic alteration of eCB metabolic components have been extensively reviewed between 2009 and 2015 [104,256,257], this section serves as an update for those employed in studying eCB metabolism in the CNS.

#### 3.3.7. 2-AG Biosynthesis and Catabolism

Multiple mouse models have been developed to study the effect of genetic alteration in the primary 2-AG metabolic enzymes (Table 14). 2-AG in the CNS is primarily synthesized via a two-step mechanism involving the conversion of phosphatidylinositol to 2-arachidonoyl-containing diacylglycerols via PLC, and subsequent deacylation via DAGL [29]. PLC is integral to multiple lipid signalling cascades and not specific to DAG metabolism; thus, research has focused on studying the effect of DAGLα/β alteration. These enzymes are encoded by the genes *Dagla* and *Daglb* (mouse chromosome 19 and 5, respectively).

Tanimura and colleagues described the first global DAGLα and DAGLβ KO models on a C57BL/6N background [258]. They employed a standard Cre-lox approach, inserting a neo cassette and lox sequences into intronic segments of these genes and crossing the resulting animals with mice constitutively expressing Cre recombinase. In the same year, Gao et al. produced a global *Dagla* loss-of-function model by inserting a GFP-neo cassette into the first exon, while taking advantage of commercial gene trap-expressing clones to produce a DAGLβ KO model as well [259]. The ES cell clone (OST195261, Lexicon Omnibank), which contains a gene trap vector in *Daglb* exon 1, gave rise to a *Daglb* null allele. Yoshino et al. also used the gene trap strategy to produce models for both *Dagla* and *-b* [260]. They employed the same *Daglb* ES clone described above, as well as another Omnibank clone (OST288027) expressing a gene trap vector in intron 4 of *Dagla*, leading to a null allele. In the same study, they described a DAGLα/β double KO, which proved that both DAGL isoforms are involved in 2-AG synthesis, although DAGLα is the prominent form in the majority of the CNS. A third group also produced a DAGLβ KO model from the OST195261 gene trap clone, only on a defined 129SvEv × C57BL/6 background [261].

Two primary models of DAGLα KO have been significant in studying the effect of 2-AG signalling in stress and anxiety. Shonesy et al. utilised an ES cell clone from the European Conditional Mouse Mutagenesis Program (ECOMM), with the targeting sequence upstream of exon 8 [262]. Jenniches and colleagues produced a model with exon 1 deleted using standard Cre-lox recombination [263].

Infrequent yet important strategies for DAGL deficiency models include region-specific gene knockout and gene silencing through RNA interference (RNAi). In the stress/anxiety context, Bluett et al. developed a DAGα^fl/fl^ transgenic animal utilizing the FLPo recombinase system and a gene trap cassette, leading to floxed exon 9 of *Dagla*. Homozygous animals were then injected with Cre-containing AAV in the basolateral amygdala to produce DAGLα KO in this region only [264]. Jain and colleagues used small interfering RNAs (siRNAs) targeting either *Dagla* or *Daglb*, determining the almost equal contribution of these two isoforms to 2-AG levels in autosynaptic (autaptic) hippocampal neuron cultures [265].
ijms-24-15829-t014_Table 14Table 142-AG enzyme genetic mouse models.Genetic BackgroundGeneral PhenotypeOutcomeReferencesC57BL/6NDAGLα^−/−^ TanimuraImproved learning habituation, increased seizure riskReduced 2-AG; abolished DSE ^1^[258]Abolished DSE ^1^ at MC-GC ^2^[266]Improved odour habituation; enhanced LTP ^3^[267]Unchanged CB1R-G protein signalling, compensatory 2-AG synthesis[268]Localization: Mostly post-synaptic[269]C57BL/6DAGLα^−/−^ GaoImpaired spatial learning and memory80% 2-AG reduction; impaired synaptic plasticity; reduced neurogenesis[259]Main contribution to brain 2-AG and eicosanoid synthesis[261]Decreased 2-AG; decreased LTD ^4^; impaired learning and memory[270]2-AG signals preferentially to neurons in short-distance synaptic regulation[34]UnspecifiedDAGLα-KO^Lex^; gene trap No overt phenotype reportedReduced 2-AG; small AEA reduction; abolished DSI ^5^[260]C57BL/6NDAGLα^−/−^Sex-specific pro-anxiety and anhedoniaCB1R-dependent anxiety; anhedonia[262]C57BL/6JDAGLα^−/−^Anxiety and fear similar to CB1R-KOIncreased fear, anxiety; loss of maternal care[263]UnspecifiedDAGLα^fl/fl^Increased susceptibility to post-traumatic stressDecreased stress resilience (AAV ^6^ directed in basolateral amygdala)[264]129SvEv × C57BL/6DAGLβ-GT^Lex^; gene trapReduced macrophage response, reduced hepatic eCB 50% 2-AG reduction; impaired synaptic plasticity; reduced neurogenesis[259]Negligible contribution to brain 2-AG and eicosanoid synthesis[261]C57BL/6NDAGLβ^−/−^No overt phenotype reportedNormal 2-AG; DSE ^1^ normal[258]UnspecifiedDAGLβ-GT^Lex^; gene trap No overt phenotype reportedNormal 2-AG; small AEA reduction[260]UnspecifiedDAGLα-GT^Lex^ × DAGLβ-GT^Lex^
Greater 2-AG reduction than single KOGreater 2-AG reduction than single KO[260]UnspecifiedRNAi DAGLα/βNo overt phenotype reportedEqual DAGLα/β contribution to autaptic CA1/3 neuron 2-AG levels[265]129SvEv × C57BL/6JMAGL-GT^Lex^; gene trap Enhanced learning, analgesic CB1R agonist toleranceCB1R desensitization; altered synaptic plasticity[271]Altered synaptic plasticity; enhanced LTD ^4^; enhanced learning[272]CB1R desensitization; prolonged climbing fibre DSE ^1^[273]C57BL6/NtacMAGL^−/−^CB1R agonist tolerance, analgesic toleranceCB1R desensitization; lack of characteristic CB1R effects[274]C57BL/6MAGL^−/−^ TaschlerCB1R agonist tolerance, anxious and obsessive-compulsive behaviourIncreased 2-AG; CB1R agonist tolerance; impaired lipolysis [275]CB1R desensitization in all regions; compensatory serine hydrolase activity[276]CB1R desensitization and disturbed E/I ratio in limbic pathways; stress-like cannabimimetic behaviour [277]C57BL/6NMAGL^−/−^ UchigashimaCB1R agonist tolerance, analgesic toleranceLow MAGL expression in MC-GC ^2^ spines, mostly astrocytic[266]UnspecifiedGluN2C:MAGL^−/−^No overt phenotype reportedProlonged 2-AG signalling, but less than total KO[278]C57BL/6MAGL^lox/lox^Floxed animals with no overt phenotype. Specific effect for neuronal, astrocytic cKO. No effect in microglial cKONeuron and astrocyte MAGL coordinate CB1R signalling termination; neuron-2-AG/astrocyte-AA ^7^ shuttle. [40]Neuron and astrocyte MAGL both contribute to terminate synaptic eCB signalling; differential effects in different fibre types/synaptic events[38]Neuron and astrocyte MAGL both contribute to terminate synaptic eCB signalling; effect restricted to molecular layer[39]Increased 2-AG; decreased AA ^7^ and PGE/F ^8^ (global KO)[85]C57BL/6NMAGL^flox/flox^Decreased microglial inflammatory responseCell-type-specific change in gene expression, astrocytic 2-AG promotes immune vigilance in microglia[279]C57BL6/JCaMKII:MAGL^Tg^Lean and hypothermic response to hypercaloric dietMAGL overexpression; decreased 2-AG correlated to weight loss and hyperthermia[280]C57BL6/JGFAP:MAGL^−/−^Decreased microglial inflammatory responseReduced neuroinflammation independent of CB1R and total 2-AG[281]DSE ^1^: depolarization-induced suppression of excitation; MC-GC ^2^: mossy cell-granule cell; LTP ^3^: long-term potentiation; LTD ^4^: long-term depression; DSI ^5^: depolarization-induced suppression of inhibition; AAV ^6^: adeno-associated viruses; AA ^7^: arachidonic acid; PGE/F ^8^: prostaglandin E and F subtypes.


The majority of findings significant to CNS therapeutic targets have resulted from models of inhibited 2-AG catabolism (Table 14)—primarily with knockdown of MAGL, which is responsible for around 85% of 2-AG clearance [253,282]. This presumably produces a “dual hit”, since the beneficial effect of 2-AG-mediated CBR signalling is amplified while the production of pro-inflammatory eicosanoids from its metabolic product (arachidonate; AA) is simultaneously diminished [32,85]. MAGL is encoded by the *Mgll* gene, located on chromosome 6 (mouse).

Schlosburg and colleagues were the first to produce a model of MAGL-KO utilizing a Lexicon Omnibank gene trap clone, with the cassette downstream of exon 3, that resulted in complete loss of function [271]. Uchigashima et al. described a targeted *Mgll* deletion through Cre-lox recombination [266]. The loxP sites were inserted to flank exon 3, and the resulting MAGL-floxed animals were mated to a germline Cre-expressing strain to produce a total KO. Taschler et al. utilised a similar strategy to produce a model that lacks exon 3 and 4 of *Mgll* [275].

As all these models showed increased brain 2-AG levels with promising neuroprotective and pro-plasticity signs, several cell-type-specific MAGL-KO models have been produced to elucidate the contribution of 2-AG metabolism in various neural contexts. Tanimura and colleagues produced a model with MAGL-KO specifically in cerebellar granule cells (CGS), utilizing the floxed strain from Uchigashima crossed to a CGS-specific-promotor-driven Cre strain [278]. Viader et al. produced conditional KO strains that have been extensively used in recent studies [40]. These models were produced with loxP sites introduced at the flanks of the catalytic *Mgll* subunit (exon 4), and cross breeding the resulting MAGL^lox/lox^ animals with neuron-, astrocyte-, and microglia-specific-promotor-driven Cre-expressing animals. Grabner et al. used a similar strategy to that of Taschler to produce MAGL^lox/lox^ animals, followed by breeding with GFAP-Cre animals to produce astrocyte-specific KO animals [281].

While most of the neuronal and astrocytic 2-AG catabolism is driven by MAGL, microglia and other distinct cell types preferably utilise ABHD enzymes, of which ABHD6 has received the most attention in the CNS context. In that regard, Zimmer’s lab was the first to describe a whole-body ABHD6-KO [283], using floxed ABHD6 ES clones from the European Conditional Mouse Mutagenesis Program (HEPD0651_8_C07). As most studies utilizing ABHD6-KO animals are in the context of adipose tissue, insulin regulation, and obesity, we could not identify other CNS-specific model studies at the time of writing. However, AAV-targeted ABHD6-KO and MAGL overexpression in the ventromedial hypothalamus has been shown to control the central signals that affect feeding responses and thermogenesis in a diet-induced obesity mouse model, thus establishing the neurological relevance of this model [33].

#### 3.3.8. AEA Synthesis and Catabolism

The main proposed pathway for AEA synthesis involves a two-step mechanism, wherein AA-conjugated phosphatidylethanolamines are converted to NAPE, and subsequently hydrolysed by NAPE-PLD, releasing AEA. The five-exon *Napepld* transcript is located on mouse chromosome 5 [41]. Several genetic models of NAPE-PLD-KO have been developed and employed to elucidate its function in a CNS and CNS disorder context (Table 15), with the most foundational models extensively reviewed elsewhere [104].

The Cravatt lab was the first to generate a whole-body NAPE-PLD-KO model, using a homologous recombination strategy to remove exon 4 that contains the catalytically active site [284]. Palmiter’s group followed with a global NAPE-PLD-KO model, produced by deleting exon 3 with a standard Cre-lox system [285]. The third foundational model was produced by Tsuboi et al., utilizing an almost identical system to knock out exon 3, albeit on a more homogenous background (C57BL6 vs. C57BL6 x 129SV in the Palmiter model) [252]. A CRISPR strategy for NAPE-PLD-KO in macrophages was recently shown in intestinal epithelial cells but has yet to be translated to an in vivo model [286].

The above models all showed a less-than-expected AEA decrease, implicating significant contribution from the other hypothesized AEA biosynthetic pathways. Thus, the Cravatt laboratory explored the contribution of GDE1 (encoded by *Gde1*; chromosome 7), an enzyme driving AEA synthesis from GP-NAPE derivatives. A GDE1-KO model was produced by removing exon 2. They also produced a double KO model resulting from a cross of GDE1-KO with the Cravatt NAPE-PLD-KO line, to explore their respective and combined contribution to AEA synthesis [287]. The same group also created a global KO model for ABHD4, another enzyme that can drive AEA synthesis from lyso-NAPE [288]. The model was created with a neo cassette replacement of exon 3 and 4, utilizing standard homologous recombination. Regarding NAPE production, a specific PLA (cytosolic PLA2ε) has recently been identified as a likely candidate for catalysing this reaction. However, recent models of cPLA2ε KO do not show significant decreases in brain NAPE or AEA levels and evidence-significant off-target reactions [289,290]. The controversial results from these models indicate that several as-of-yet uncharacterised pathways contribute to AEA tone, highlighting the functional redundancy in NAE biosynthetic pathways.
ijms-24-15829-t015_Table 15Table 15AEA enzyme genetic mouse models.Genetic BackgroundGeneral PhenotypeOutcomeReferences129SvJ × C57BL/6NAPE-PLD^−/−^ CravattHealthy and viable through lifespan, no significant AEA decreaseDecreased saturated and monounsaturated NAEs, no change in AEA[284]Subcellular localization: Preferentially in dendrites[269]Lower AEA; higher DHA/DHEA; significant effect of dietary fatty acids[291]AEA signals preferentially to astrocytes; astrocytic Ca^2+^ mobilization and synaptic plasticity[34]C57BL/6 × 129SVNAPE-PLD^−/−^ PalmiterHealthy and viable throughout, significant AEA decreaseDecreased AEA/NAE levels, ABDH4 as main compensatorysynthesizer[285]Region- and lipid-specific NAE/MAG alteration; decreased NAE[255]C57BL/6NAPE-PLD^−/−^ TsuboiHealthy and viable throughout, significant AEA decreaseDecreased AEA/NAE levels, compensatory synthesis pathways involved[252]Decreased AEA/NAE levels, compensatory synthesis pathways involved[292]C57BL/6GDE1^−/−^Healthy and viable through lifespan, no significant AEA decreaseUnchanged NAE/AEA levels[287]C57BL/6GDE1^−/−^ × NAPE-PLD^−/−^Healthy and viable through lifespan, no significant AEA decreaseUnchanged NAE/AEA levels[287]129SvEv × C57BL/6ABHD4-KOHealthy and viable through lifespan, no significant AEA decreaseNo significant AEA decrease; decreased GP/pNAPE [288]129SvJ × C57BL/6FAAH^−/−^Significant AEA increase, pain and inflammation resistance, super sensitivity to AEA treatmentIncreased AEA; CB1R-dependent sensitivity to AEA[293]Increased GABAergic inhibition of synaptic transmission[294]Decreased AEA transport; diffusion- and transporter-mediated systems[295]CB1R-mediated hypoalgesia[296]Increased neurogenesis (AEA-dependent)[297]Decreased post-mortem AEA/EA accumulation[298]Increased astrogliogenesis (CB1R-dependent)[199]Alternative phosphocholine-NAE metabolic route[299]Accumulation of N-acyltaurines, which act as TRPV1/4 agonists[300]Resistance to acetaminophen analgesia, TRPV1-dependant[301]NAE/NAT accumulation in CNS (primarily long-chain saturated forms)[302]Altered iLTD [303]Increased TRPV1 activity; modulated glutamate release[304]Hyperalgesia to certain types of pain; analgesia to others[305]Increased AEA and phosphamides; no changes in PG_E/F_ effect[85]129SvJ × C57BL/6FAAH^−/−^ × Eno2:FAAHWild-type pain responses, resistance to inflammationIncreased AEA in CNS; no change in periphery[306]


Catabolism of AEA is less ambiguous, with only a few enzymes shown to contribute to most of its clearance in the CNS. FAAH1 in mice, encoded by the *Faah* gene (chromosome 4), catalyses the majority of AEA hydrolysis.

Cravatt et al. produced the first genetic model of FAAH-KO, utilizing a neo cassette and recombination strategy to delete exon 1 and its upstream expression control region, creating a null allele [293]. To the best of our knowledge, all FAAH-KO studies performed to date (except those utilizing conditional or transgenic models) utilised this strain (Table 15). Knockdown of NAAA, another main catabolic enzyme of AEA, has recently been modelled using a cutting-edge CRISPR-Cas9 approach [307]; however, this model has not been applied to CNS studies to date. Regarding cell-type-specific FAAH models, Cravatt and colleagues also produced an altered version of their whole-body KO with FAAH expression re-introduced only in neurons [306]. A transgenic model was created to investigate the role of FAAH in a common model of amyotrophic lateral sclerosis (ALS), by crossing the Cravatt strain FAAH-KO mice with SOD-Tg animals [308]. The outcomes of this and other studies that applied eCB metabolic enzyme models to CNS disorder research are summarized in Table 16.

## 4. Concluding Remarks

The ability of neuroscience to meet the growing need for novel therapies is directly dependent on our ability to obtain a precise understanding of the underlying mechanisms in health and disease. It is becoming ever more important for researchers to stay updated on the molecular–genetic tools in our arsenal, as the decision of model and methodology ultimately defines the quality of knowledge we can generate towards meeting this urgent need. In few instances is this truer than in investigations regarding the ECS. Several ECS-targeting therapeutics have been approved or are in the process of approval for various neuropathologies, yet we are only now beginning to appreciate the full ubiquity and complexity of this system.

In this review, we indexed some of the most impactful tools to assess the ECS within the context of the CNS and related disorders. We reviewed new imaging tools such as STORM and GRAB_eCB2.0_ in comparison to classical techniques and summarized the best genetic models available to date. As science continuously evolves, new ligands and antibodies will keep standard imaging tools relevant, but advances in AI and computing will significantly facilitate overcoming existing limitations in techniques like MALDI, STORM, and TEM. Some genetic models for ECS investigation have been extensively utilised while others are highly disputed in the scientific community, leaving the door open for new models to arise and help fill the remaining knowledge gaps. Indeed, many questions regarding the ECS’s involvement in CNS and CNS disorders remain unanswered, which the plethora of available tools can help us address in the future.

## Figures and Tables

**Figure 1 ijms-24-15829-f001:**
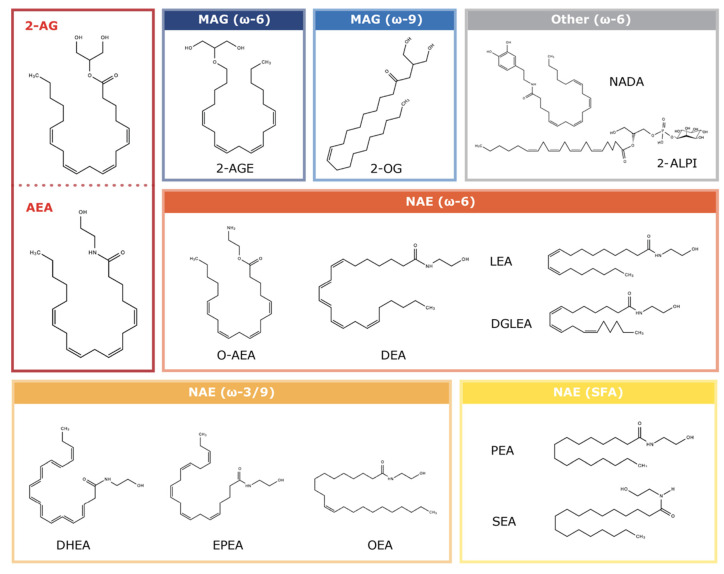
Structural relationships between known eCBs and eCB-like lipid signalling molecules. Current evidence indicates eCBs are majorly derived from the NAE and MAG signalling lipid classes, although few endogenous CB1R agonists with different backbones (NADA and 2-ALPI) are also known. The fatty acyl groups most often include omega-3/6/9 PUFA species or long-chain saturated fatty acids.

**Figure 2 ijms-24-15829-f002:**
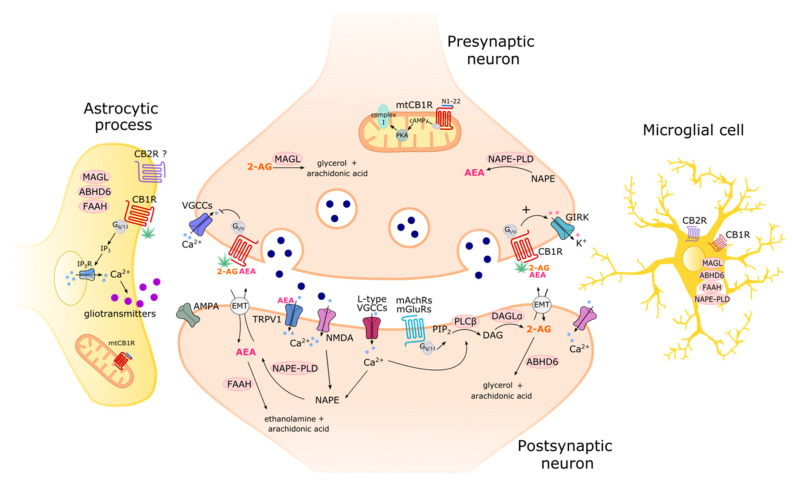
Main proposed mechanisms for eCB metabolism and their action on CBRs in neurons and glia. Neuronal activation leads to on-demand eCB production as neurotransmitters activate either voltage-gated Ca^2+^ channels (VGCCs) and NMDA receptors, or stimulation of metabotropic receptors (mGluRs and mAChRs). AEA is mainly synthesized from the enzymatic hydrolysis of the membrane precursor N-arachidonoyl phosphatidylethanolamine (NAPE) via a specific phospholipase D (NAPE-PLD). 2-AG biosynthesis starts with GPCR-activated PLCβ hydrolysing phosphatidylinositol 4,5-bisphosphate (PIP2) to inositol 1,4,5-triphosphate and diacylglycerol (DAG), which is next hydrolysed by the enzyme diacylglycerol lipase, mainly the α isoform (DAGLα). The main AEA-degrading enzyme, fatty acid amide hydrolase (FAAH), is localized at postsynaptic sites, while the main 2-AG-hydrolyzing enzyme, monoacylglycerol lipase (MAGL), is located at presynaptic axon terminals. Lesser amounts of 2-AG are also hydrolysed at postsynaptic sites by α/β-hydrolase-6 (ABHD6). eCBs (or exogenous cannabinoids like THC) stimulate presynaptic CB1R, which leads to suppressed neurotransmitter release. This effect mainly involves GPCR-dependant inhibition of presynaptic Ca^2+^ influx through VGCCs and/or an activation of inwardly rectifying K^+^ channels (GIRK). Astrocytic CB1Rs are reported to induce a rise in intracellular Ca^2+^ and increase the release of gliotransmitters such as glutamate. In the mitochondria, stimulation of mtCB1R has been shown to modulate mitochondrial respiration and metabolism, likely via the cAMP/PKA/complex I pathway. ?, CB2R presence in astrocytes remains questionable.

**Figure 3 ijms-24-15829-f003:**
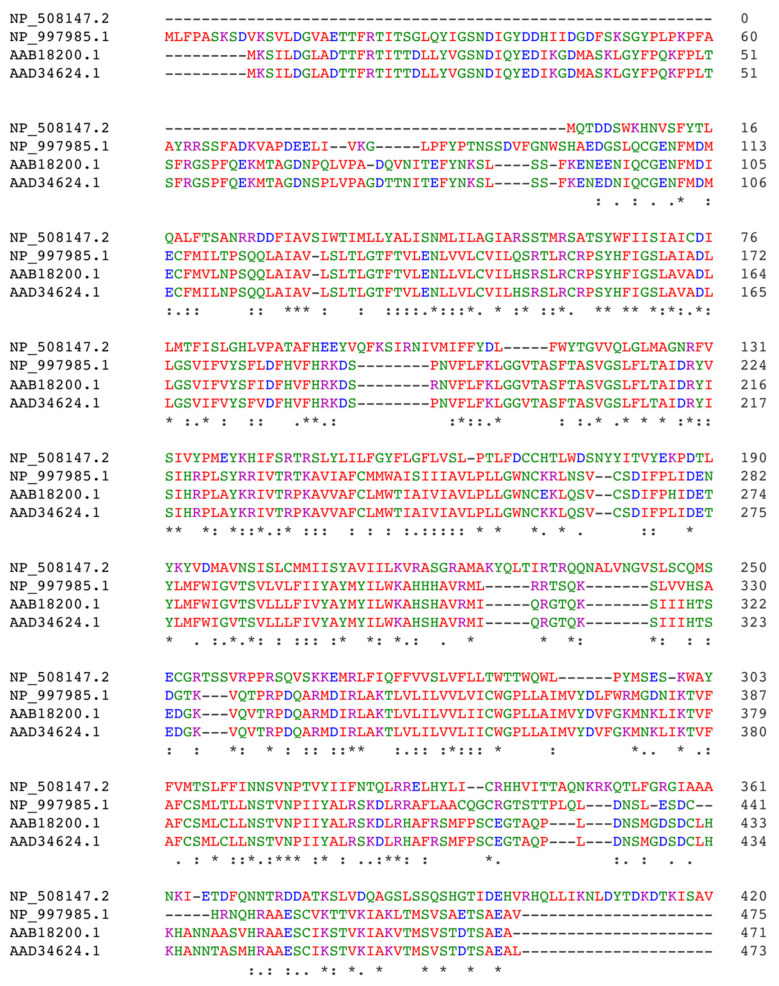
Alignment of CB1R protein sequences from *C. elegans*, zebrafish, mouse, and human. The alignment was performed using Clustal Omega software 1.2.4 (https://www.ebi.ac.uk/ accessed on 11 September 2023) with the following sequences: NP_508147.2 (*C. elegans*), NP_997985.1 (*Danio rerio*), AAD34624.1 (*Mus musculus*), AAB18200.1 (*Homo sapiens*). Asterisk (*) fully conservated, colon (:) more conservative substitution, period (.) less conservative substitution.

**Table 1 ijms-24-15829-t001:** Key components of the ECS.

**Cannabinoid Receptors**
CB1R, CB2R, GPR55, GPR18, GPR119, TRPs, PPARs
**Endocannabinoids**
2-AG, AEA, PEA, DHEA, EPEA, OEA, NADA, 2-AGE, O-AEA
**Enzymes**
Biosynthesis: DAGLα/β, NAPE-PLD, ABHD4, GDE1, PLC, PLA, PTN22, SHIP1 Degradation: FAAH, MAGL, NAAA, ABHD6/12, COX-2, CYP2-4, 12/15-LO Transport process: EMT, FABP5/7, SCP2/x

**Table 2 ijms-24-15829-t002:** Overview of techniques used to image the components of the ECS in the CNS, including their technical advantages, disadvantages, and those ECS components that have been studied with each tool.

Technique	Advantages	Disadvantages	ECS Components
Radioligands			
Radioactive ligands coupled with PET ^1^, autoradiography, radioligand binding assays	Non-invasive in vivo or ex vivo tool Target visualization Pharmacodynamic and pharmacokinetic data No need to validate antibody Alternative to immunofluorescence High tissue penetration and can be used for whole-body imaging	Radiation exposure Relatively expensive Low spatial resolution and long acquisition time	CB1R, CB2R, enzymes
Optical microscopy			
Fluorescence light microscopy (Immunocytochemistry and immunohistochemistry)	Can be used with low magnifications Low cost High sensitivity Good resolution High simplicity High speed Can be used to elucidate distributions from organ to subcellular level	Lower resolution compared to more advanced techniques Limited tissue penetration Potential interference of endogenous molecules Requires antibody validation	CB1R, CB2R, enzymes
FRET ^2^	Very high resolution Relatively low cost compared to electron microscopy Can be used to study protein interactions	Lack of appropriate labelling for intracellular proteins Target molecules must be close to each other	CB1R, CB2R
Electron microscopy			
TEM ^3^	Very high resolution Element and structural compound data retrieval	Relatively expensive Laborious sample preparation High risk of contamination from sample preparation processes Time consuming	CB1R, CB2R, enzymes
Super-resolution microscopy			
STORM ^4^	High lateral, axial, and spatial resolution 3D-STORM Can be used to study target distributions at nanoscale	Large data processing Low speed Requires specialized fluorophores	CB1R
Mass spectrometry imaging (MSI)			
MALDI ^5^	Specimen detection with high sensitivity and specificity	No spatiotemporal resolution Limited quantitative power	eCB
Bioengineered sensors			
GRAB_eCB2.0_ ^6^	Higher spatiotemporal resolution than MSI	Very recent technique that still requires further investigation	eCB

PET ^1^: positron emission tomography; FRET ^2^: fluorescence resonance energy transfer; TEM ^3^: transmission electron microscopy; STORM ^4^: Stochastic Optical Localization Microscopy; MALDI ^5^: matrix-assisted laser desorption/ionization; GRAB_eCB2.0_
^6^: GPCR-activation-based eCB sensor.

**Table 5 ijms-24-15829-t005:** Imaging tools used to study eCBs in the CNS, with particular focus on their respective technical advantages and disadvantages.

Technique	Advantages	Disadvantages	References
Mass spectrometry imaging			
MALDI ^1^	Specimen detection with sensitivity and specificity	Low spatiotemporal resolution Limited quantitative power	[88]
Bioengineered sensors			
GRAB_eCB2.0_ ^2^	Higher spatiotemporal resolution than MSI	Unable to distinguish 2-AG from AEA	[89]

MALDI ^1^: matrix-assisted laser desorption/ionization; GRAB_eCB2.0_
^2^: GPCR-activation-based eCB sensor.

**Table 7 ijms-24-15829-t007:** Summary of zebrafish genetic models related to the ECS.

Zebrafish Stage	Genetic Approach	Technique	Outcome	References
Juvenile	*cnr1* KO (total)	Morpholino	Defects in axonal growth and fasciculation	[128]
Juvenile	*cnr1* and *cnr2* KO (total)	Morpholino	CART-3 ^1^ expression and yolk sac size	[132]
Juvenile	*cnr1* KO (total)	Morpholino	Decreased locomotor activity and suppression of feeding behaviour	[133]
Juvenile Adult	*cnr1* overexpression (hepatic)	Tet^off^ transgenic system	Loss of lipid accumulation	[131]
Juvenile	*cnr2* KO (total)	CoDA ZFN ^2^	Regulation of leukocyte migration	[134]
Juvenile	*cnr2* KO (total)	Morpholino	Reduced *runx1* expression and decreased HSCs ^3^	[135]
Juvenile	*cnr1* KO (total)	Morpholino	Reduced microRNA dre-let-7d levels	[136]
Juvenile Adult	*cnr1* and *cnr2* KO	Morpholino TALEN ^4^	Smaller livers with fewer hepatocytes, reduced liver-specific gene expression and proliferation	[134]
Juvenile Adult	*cnrip1a* and *cnrip1b* (total)	CRISPR ^5^/Cas9	No phenotype is detected, fish lacking these genes both maternally and zygotically are viable.	[137]
Juvenile	*cnr2* KO (total)	CRISPR/Cas9	Mutants show an anxiety-like behaviour: they show an altered PDR ^6^ and decreased CO ^7^	[138]
Juvenile	*cnr1* KO (total)	Morpholino	Reduced number of GnRH3 neurons, fibre misrouting, and altered fasciculation	[139]

CART-3 ^1^_:_ cocaine- and amphetamine-related transcript; CoDA ZFN ^2^: context-dependent assembly zinc-finger nuclease; HSC ^3^: hematopoietic stem cells; TALEN ^4^: transcription-activator-like effector nuclease; CRIPSR ^5^: clustered regularly interspaced short palindromic repeats; PDR ^6^: photo-dependent swimming responses; CO ^7^: centre occupancy.

**Table 16 ijms-24-15829-t016:** eCB metabolic enzyme genetic mouse models in CNS disorders.

Disorder	Model	Outcome	References
ALS ^1^	FAAH^−/−^ × SOD1-Tg	Attenuated symptoms; no increase in lifespan (CB1R-independent)	[308]
Diet-induced obesity	ABHD6^−/−^	No changes in total MAG content	[283]
	ABHD6^−/−^	Increased 2-AG/CB1R signalling drives weight gain and hypothermia	[33]
	CaMKII:MAGL^Tg^	MAGL overexpression; decreased 2-AG correlated to weight loss and hyperthermia	[280]
EAE ^2^ model of MS ^3^	FAAH^−/−^	Enhanced recovery/remission from EAE ^2^	[309]
Kainate-induced epilepsy	FAAH^−/−^	Increased seizure susceptibility	[310]
	DAGLα^−/−^ Tanimura	Decreased CB1R DSI ^4^; increased seizure incidence	[311]
Pain and neuroinflammation (LPS ^5^ induced)	GFAP:MAGL^−/−^	Reduced neuroinflammation independent of CB1R and total 2-AG	[281]
	DAGLα^−/−^ Gao	No effect on pain sensitization	[312]
	DAGLβ-GT^Lex^; gene trap	Reduced pain sensitization	[312]
Parkinson’s disease	MAGL-GT^Lex^; gene trap	Reduced neuroinflammation; neuroprotection; prostaglandin and not ECS related	[32]
	DAGLβ-GT^Lex^; gene trap	DAGLβ is main 2-AG synthesizer in SN ^6^; contributes to disease progression in mice and rare-variant patients	[313]
Stress + anxiety	DAGLα^fl/fl^	Decreased stress resilience (BLA ^7^ AAV ^8^-directed)	[264]
	DAGLα^fl/fl^	Reduced stress behaviour	[314]
	DAGLα^−/−^	Increased anxiety; anhedonia	[262]
	DAGLα^−/−^	Increased fear, anxiety; loss of maternal care	[263]
	FAAH^−/−^	Reduced anxiety	[315]
Substance use disorders	FAAH^−/−^	Increased ethanol consumption and preference	[316]
	FAAH^−/−^	Increased alcohol sensitivity and withdrawal	[317]
	FAAH^−/−^	Reduced morphine withdrawal	[318]
TBI ^9^	DAGLβ-GT^Lex^; gene trap	Sex-dependant increase in survival; attenuated sphingolipid TBI ^9^ markers	[319]
	GFAP:MAGL^flox/flox^	Reduced neuroinflammation, neuroprotection, CB1R-PPARγ-dependent	[320]

ALS ^1^: amyotrophic lateral sclerosis; EAE ^2^: experimental autoimmune encephalomyelitis; MS ^3^: multiple sclerosis; DSI ^4^: depolarization-induced suppression of inhibition; LPS ^5^: lipopolysaccharides; SN ^6^: substantia nigra; BLA ^7^: basolateral amygdala; AAV ^8^: adeno-associated virus; TBI ^9^: traumatic brain injury.

## Data Availability

No new data were created in this study except the alignment of CB1R, which is contained within the article (Figure 3).

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
