# Peer review of "Imaging and Genetic Tools for the Investigation of the Endocannabinoid System in the CNS"

_ijms, 2023, doi:10.3390/ijms242115829_

Round 1

Reviewer 1 Report

Comments and Suggestions for Authors

The manuscript is impressively well-written and meticulously organized. The authors have presented their research coherently and systematically, making it comprehensive and accessible for the readers. The structure flows logically from section to section, demonstrating thoroughness in the topic discussed. Overall, it stands as a commendable piece of academic work.

1: The abstract is generally well-structured and clearly presents the paper's main topics. It could benefit from a few organizational tweaks to make the flow smoother.

2: Please ensure all statements requiring citations have appropriate references. Additionally, ensure that all cited works in the text are in the bibliography.

3: It would enhance clarity if figures and tables were sequentially labeled and appropriately referenced within the text. Ensure that each figure and table has a descriptive caption that provides context to the reader.

4: Ensure consistent use of terminology throughout the manuscript. Avoid switching between terms that refer to the same concept.

5: While the content is scientifically rigorous, some grammatical errors could impede readability. Consider thorough proofreading or utilizing a professional editing service to polish the language.

Comments on the Quality of English Language

Minor editing of the English language required

Author Response

We appreciate the reviewers for dedicating their time to a thorough evaluation of our manuscript and for offering constructive feedback. We are particularly pleased with their positive assessments of the importance and quality of our work. We have carefully considered their suggestions and implemented several revisions, resulting in a substantially improved version of the manuscript. Specific responses to the reviewers’ comments are provided below and the main modifications have been highlighted in yellow in the revised version of the manuscript.

We would like to point out that many of the reviewers’ concerns relate to typography (super- and subscripts in abbreviations and special characters) were an unintentional consequence of software errors in the review process. As we rely on a reference manager that is incompatible with the MDPI template, we submit the current manuscript in a plain Microsoft Word document, which the MDPI editorial team copies to the template before being sent out to reviewers. We suspect these special characters are lost in this transfer of formats, and we will do all in our power to ensure this oversight is not repeated in the following submission.

While the reviewers generally expressed satisfaction with the manuscript's English language quality, they highlighted specific cases of unclear language use. Therefore, any modifications in the manuscript not explicitly addressed in response to specific reviewer comments were made to further simplify and clarify the language.

Reviewer 1

1: The abstract is generally well-structured and clearly presents the paper's main topics. It could benefit from a few organizational tweaks to make the flow smoother.

We thank the reviewer for this comment. We have introduced some tweaks as suggested to improve the abstract’s flow.

2: Please ensure all statements requiring citations have appropriate references. Additionally, ensure that all cited works in the text are in the bibliography.

We have ensured that every statement is correctly cited and two authors independently verified the integrity of the bibliography and in-text citations.

3: It would enhance clarity if figures and tables were sequentially labelled and appropriately referenced within the text. Ensure that each figure and table has a descriptive caption that provides context to the reader.

We thank the reviewer for again pointing out a technical error that derived from document transfer between authors and the editorial team. We have ensured all figures and tables are sequentially and descriptively labelled and that they are referenced to in the text in a consistent manner.

4: Ensure consistent use of terminology throughout the manuscript. Avoid switching between terms that refer to the same concept.

Again, thanks to the reviewer for the helpful tip. We have standardized our use of terminology and spelling of abbreviations throughout the manuscript.

5: While the content is scientifically rigorous, some grammatical errors could impede readability. Consider thorough proofreading or utilizing a professional editing service to polish the language.

We thank the reviewer for his/her honest feedback. Two of the four authors have carefully proofread the manuscript again after all edits were completed. We did not deem it necessary to hire professional editing services in the short time frame that was given for this rebuttal as the feedback on our use of language was generally positive and the main proofreading author is a native English speaker.

Reviewer 2 Report

Comments and Suggestions for Authors

The authors reviewed the available literature about imaging and genetic tools to investigate the endocannabinoid system's function in the central nervous system. The authors collected a remarkable amount of information about state-of-the-art techniques and tools comparing pros and cons for each one. They cited the appropriate references for every topic. The manuscript is well-written and suitable for publication in IJMS.

 Minor comments:

-check CB1R and CB1R for consistency (in both main text and legends; same for CB2R)

-fix the weird symbols of PPAR, PLC, and synuclein (in both main text and legends)

Author Response

We appreciate the reviewers for dedicating their time to a thorough evaluation of our manuscript and for offering constructive feedback. We are particularly pleased with their positive assessments of the importance and quality of our work. We have carefully considered their suggestions and implemented several revisions, resulting in a substantially improved version of the manuscript. Specific responses to the reviewers’ comments are provided below and the main modifications have been highlighted in yellow in the revised version of the manuscript.

We would like to point out that many of the reviewers’ concerns relate to typography (super- and subscripts in abbreviations and special characters) were an unintentional consequence of software errors in the review process. As we rely on a reference manager that is incompatible with the MDPI template, we submit the current manuscript in a plain Microsoft Word document, which the MDPI editorial team copies to the template before being sent out to reviewers. We suspect these special characters are lost in this transfer of formats, and we will do all in our power to ensure this oversight is not repeated in the following submission.

While the reviewers generally expressed satisfaction with the manuscript's English language quality, they highlighted specific cases of unclear language use. Therefore, any modifications in the manuscript not explicitly addressed in response to specific reviewer comments were made to further simplify and clarify the language.

Reviewer 2

  1. Check CB1R and CB1R for consistency (in both main text and legends; same for CB2R)

We have now changed all the “CB1R” and “CB2R” into CB1R and CB2R. This oversight was due to an error in text transfer to the MDPI template after our manuscript submission as described in the opening statements of this letter. We have discussed this with the MDPI editorial team and from both parties will ensure this technical problem is solved before the reviewers receive the corrected manuscript.

  1. Fix the weird symbols of PPAR, PLC, and synuclein (in both main text and legends)

We again thank the reviewer for his/her vigilance, this is another issue derived from the formatting problems described above, where all the Greek letters were changed. We have fixed the problem and will ensure this technical problem is solved before the reviewers receive the corrected manuscript.

Reviewer 3 Report

Comments and Suggestions for Authors

This is a nice review of the molecular composition of the endocannabinoid system, the tools used for studying these molecules, and the knockout animals available to study their functions. Tables 10-16 in particular provide a summary of the available mouse models and their phenotypes; these will serve as a useful reference for many readers.

I only have some minor suggestions and comments to add.

1.     Line 124: “on-demand” ECB production: this question is still controversial, and will likely remain to be until the molecular mechanism of endocannabinoid transport is revealed. While solving this problem is outside the scope of this review, a caveat sentence is warranted. It is nearly impossible to differentiate between on-demand synthesis, and the controlled release from a constantly replenished pool.

2.     Figure 1 of the chemical structures is not particularly useful or relevant for the review, especially that only 2-AG and AEA are discussed in further detail. Are there any interesting features of the chemical structures of endocannabinoids, or a relevant grouping of endocannabinoids, that this figure should highlight?

3.     Line 148. What are eicosanoids and eicosanoid metabolism? If these are mentioned, a brief explanation would be helpful.

4.     Line 163. The “is usually cited in confirmation of” language is cryptic. Is this to suggest that there are other endocannabinoid metabolic routes that are not related to retrograde synaptic regulation? Or?

5.     Line 259-262, The description of electron microscopy is vague and focuses on features that are not really scientifically relevant.

a.     What does it mean the electron microscope is producing images of “high quality”? Arguably, low-contrast and grainy electron microscopic images of poorly preserved neuropil look quite unappealing compared to beautiful color fluorescence images of elegant neurons. How is “image quality” different than the greater resolution also (correctly) mentioned?

b.     Electron microscopy is repeatedly described as being “very expensive”. It definitely is; however, STORM or PET are expensive too, and in some institutions one or more of these methods are available to users at a reasonable price through core facilities. 

6.     Tables 2-3. The listed advantages and disadvantages focus too much on the practical aspects of the discussed methods and less on the types of questions that can be addressed with one or the other approach.

7.     Line 291-300. The section in CB1 immunostaining is superficial, and highlights a seemingly arbitrary choice of publications (without reviewing some classic studies on CB1 immunolocalization, or the antibodies that made these available). N-terminal antibody is mentioned, but relative advantages of N- and C-terminal targeting CB1 antibodies are not discussed (for live cell imaging or electron microscopy, respectively).

8.     Line 442. Name of van der Stelt misspelled.

9.     Line 458. Presynaptic and postsynaptic dendrites: maybe use neurites, or axons and dendrites.

10.  Line 466: which invertebrate groups have both CB1 and CB2? The C. elegans sections reads as there’s no evidence for a CB2 homolog.

11.  What is the proposed function of endocannabinoids in the worm? Was there an evolution of the roles of endocannabinoid signaling from a basic cellular metabolic towards a synaptic role? If so, when during phylogeny these changes occurred?

12.  Line 521. What does the zebrafish’s genomic similarity to mammals mean? Does this just mean that fish are more similar to mammals compared to invertebrates (which is trivial), or that zebrafish has some interesting genomic structure that makes them better models than other fish?

13.  Line 538. A brief description of the morpholino gene silencing method would be useful.

14.  Line 538 says the first zebrafish CB1 KO was ref. 91, line 557 says it was ref. 81.

15.  Line 561-563: the description of ref. 111 reads like a basic control experiment (gene knockdown reduces gene expression). Not necessary to include in a review.

16.  Line 574. What is CART? How is this relevant to endocannabinoids, what should the reader take home about this?

17.  Line 613. Same question for runx1.

18.  Line 599. What is the meaning of the listed cnr2 target regions?

19.  Line 619, also 745: Anxiety-like behaviors. Can the authors discuss what is the proposed mechanism of this, if CB2R is not expressed in the brain?

Author Response

We appreciate the reviewers for dedicating their time to a thorough evaluation of our manuscript and for offering constructive feedback. We are particularly pleased with their positive assessments of the importance and quality of our work. We have carefully considered their suggestions and implemented several revisions, resulting in a substantially improved version of the manuscript. Specific responses to the reviewers’ comments are provided below and the main modifications have been highlighted in yellow in the revised version of the manuscript.

We would like to point out that many of the reviewers’ concerns relate to typography (super- and subscripts in abbreviations and special characters) were an unintentional consequence of software errors in the review process. As we rely on a reference manager that is incompatible with the MDPI template, we submit the current manuscript in a plain Microsoft Word document, which the MDPI editorial team copies to the template before being sent out to reviewers. We suspect these special characters are lost in this transfer of formats, and we will do all in our power to ensure this oversight is not repeated in the following submission.

While the reviewers generally expressed satisfaction with the manuscript's English language quality, they highlighted specific cases of unclear language use. Therefore, any modifications in the manuscript not explicitly addressed in response to specific reviewer comments were made to further simplify and clarify the language.

Reviewer 3

  1. Line 124: “on-demand” ECB production: this question is still controversial, and will likely remain to be until the molecular mechanism of endocannabinoid transport is revealed. While solving this problem is outside the scope of this review, a caveat sentence is warranted. It is nearly impossible to differentiate between on-demand synthesis, and the controlled release from a constantly replenished pool.

We agree that the mechanisms behind the on-demand synthesis are far from being clear and the topic is quite controversial. Thus, we have edited the passage in question to clearly state the caveats made by the reviewer.

  1. Figure 1 of the chemical structures is not particularly useful or relevant for the review, especially that only 2-AG and AEA are discussed in further detail. Are there any interesting features of the chemical structures of endocannabinoids, or a relevant grouping of endocannabinoids, that this figure should highlight?

We agree with the reviewer’s concern that the first version of Figure 1 was not particularly useful in the context of the review. Two major changes to this figure have been now made to improve this: 1) Grouping of the endocannabinoids according to their chemical backbone and fatty acid conjugates 2) addition of novel and lesser-described endocannabinoids. The relevant text has been updated with this information, as well as the figure caption. Overall, we believe this figure now provides a level of novelty and relevance, since it describes their structural relationship of other eCBs to the main cannabinoids (2-AG and AEA) that are the focus of the review.

  1. Line 148. What are eicosanoids and eicosanoid metabolism? If these are mentioned, a brief explanation would be helpful.

We agree with the reviewer’s suggestion as it would increase the value of this manuscript to those readers unfamiliar with the complex field of bioactive lipids. In the last sentences of the introductory paragraph in section 1.3., we have now included a brief explanation of eicosanoids, what they are, which process are involved in, their metabolism and how all these facts are relevant to the enzymes of the ECS.

  1. Line 163. The “is usually cited in confirmation of” language is cryptic. Is this to suggest that there are other endocannabinoid metabolic routes that are not related to retrograde synaptic regulation? Or?

We admit that the use of language used here was too cryptic. It was not meant to implicate any other metabolic routes, only emphasize the fact that the distribution of the involved enzymes is in line with the retrograde synaptic signaling model. The relevant sentence has been now changed accordingly to avoid this ambiguity.

  1. Line 259-262, The description of electron microscopy is vague and focuses on features that are not really scientifically relevant.

We thank the reviewer for his/her insight. This section has been updated to highlight more relevant information on the technique, according to the specific problems detailed below:

  1. What does it mean the electron microscope is producing images of “high quality”? Arguably, low-contrast and grainy electron microscopic images of poorly preserved neuropil look quite unappealing compared to beautiful color fluorescence images of elegant neurons. How is “image quality” different than the greater resolution also (correctly) mentioned?

We have now changed the text and tables to avoid misunderstandings. Very high resolution is stated in the table and text as an advantage of TEM, but that issues with sample preparation are more likely.

  1. Electron microscopy is repeatedly described as being “very expensive”. It definitely is; however, STORM or PET are expensive too, and in some institutions one or more of these methods are available to users at a reasonable price through core facilities. 

We have now changed the text referring to Electron microscopy as a technique more expensive relatively to optical microscopy.

  1. Tables 2-3. The listed advantages and disadvantages focus too much on the practical aspects of the discussed methods and less on the types of questions that can be addressed with one or the other approach.

We thank the reviewer for his/her concern. The purpose of the tables was originally to summarize these techniques shortly with a specific focus on their practical advantages and disadvantages. Furthermore, the text related to tables 2 and 3 delve further into the types of research questions that have successfully been investigated with each technique. Nevertheless, we have introduced two corrections to consolidate our approach according to the reviewer’s concern:

1) We have updated these tables, adding the potential applications of these techniques, when applicable, and 2) renamed the tables to explicitly state their focus to be on the practical/technical aspects.

  1. Line 291-300. The section in CB1 immunostaining is superficial, and highlights a seemingly arbitrary choice of publications (without reviewing some classic studies on CB1 immunolocalization, or the antibodies that made these available). N-terminal antibody is mentioned, but relative advantages of N- and C-terminal targeting CB1 antibodies are not discussed (for live cell imaging or electron microscopy, respectively).

We thank the reviewer for his/her expert insights to this section. We have now included and cited key papers regarding the first assessments of CB1R distribution using both N-terminal and C-terminal tagged antibodies in this section.

However, we believe that a systematic digression on N-terminal and C-terminal antibodies is out of the scope of this review.  Therefore, we include a caveat sentence at the end of this paragraph to make the reader aware of the differential usefulness of N- or C-terminal targeting antibodies and refer them to the studies cited for further detail.

  1. Line 442. Name of van der Steltmisspelled.

We are grateful to the reviewer for pointing out this small yet important spelling error that slipped through the cracks during proofreading. It has been corrected in the text.

  1. Line 458. Presynaptic and postsynaptic dendrites: maybe use neurites, or axons and dendrites.

We have taken note of this confusing phrasing and changed the respective line to read “neurites close to the synapse”. The main point here was to emphasize the difference in NAPE-PLD localization compared to the seemingly more somatic distribution of FAAH.

  1. Line 466: which invertebrate groups have both CB1 and CB2? The C. elegans sections reads as there’s no evidence for a CB2 homolog.

We again apologize for the unclear language here, as the sentence was not meant to be interpreted as both invertebrates and vertebrates having both receptors. It rather aims to state that homologs for both receptors have been found to various extents in members from either of these two main groups of animals. The relevant part of the introductory paragraph to section 3 has been simplified and rewritten to clarify this point.

  1. What is the proposed function of endocannabinoids in the worm? Was there an evolution of the roles of endocannabinoid signaling from a basic cellular metabolic towards a synaptic role? If so, when during phylogeny these changes occurred?

We would like to thank the reviewer for these questions. We have summarized the basic functions in which endocannabinoids play a role in C. elegans at the end of section 3.1.1.  Included here are recent findings which show that endocannabinoids may influence several neurotransmitter circuits in a manner dependent on novel putative CBRs.  

To the best of our knowledge, there is no literature describing the evolution of cannabinoid receptors and signaling at the required level of detail to infer when and how it came to be. As this is still relatively novel information, no expert review of its evolutionary implications has been published thus far. We therefore prefer not to comment on the probable evolutionary purpose, origin and timeframe for its emergence to avoid digressing into pure speculation that is not relevant to the point made in this section.

  1. Line 521. What does the zebrafish’s genomic similarity to mammals mean? Does this just mean that fish are more similar to mammals compared to invertebrates (which is trivial), or that zebrafish has some interesting genomic structure that makes them better models than other fish?

We thank the reviewer for pointing out this discrepancy. All that was implied with this section is that the well-characterized genome and human gene orthologs in zebrafish, together with the practical advantages they share with C. elegans models, make them good models for ECS studies at present.

We have edited the last sentence of section 3.1.2. and the list of advantages in the first paragraph of section 3.2. accordingly, avoiding the ambiguous phrasing “genomic similarity to mammals”.  

  1. Line 538. A brief description of the morpholino gene silencing method would be useful.

As suggested, we have now added a short definition of morpholino technique and a foundational reference for a better understanding of this section.

  1. Line 538 says the first zebrafish CB1 KO was ref. 91, line 557 says it was ref. 81.

We thank the reviewer for pointing out this error. We have now corrected the reference citing the proper paper in both cases.

  1. Line 561-563: the description of ref. 111 reads like a basic control experiment (gene knockdown reduces gene expression). Not necessary to include in a review.

We have now excluded this basic control experiment from the text.

  1. Line 574. What is CART? How is this relevant to endocannabinoids, what should the reader take home about this?

We thank the reviewer for the opportunity to provide further context on this interesting transcript. We have now included sentences in this section to briefly define CART, its importance regarding endocannabinoids, as well as the interaction between them. The significance of the findings has been contextualized in appetite regulation and the endocannabinoid field.

  1. Line 613. Same question for runx1.

As for the previous question, we have now briefly defined runx1 and described its potential role and importance in the ECS context.

  1. Line 599. What is the meaning of the listed cnr2 target regions?

The two listed sequences correspond to those that codify for the two exons of cnr2 gene. Whereas morpholinos targeting both exons have been developed for cnr2, morpholinos of cnr1 targeting only exon 2 have been developed. Therefore, we listed the cnr2 target regions to take in consideration the differences that may arise because of targeting different exons. We have now included a sentence at the end of the first paragraph of section 3.3.4. to explicitly make the reader aware thereof.

  1. Line 619, also 745: Anxiety-like behaviors. Can the authors discuss what is the proposed mechanism of this, if CB2R is not expressed in the brain?

We thank the reviewer for pointing out this nuanced inconsistency in evidence. We have included a sentence at the end of the referenced section with an additional citation to make this clear to the reader.

Reviewer 4 Report

Comments and Suggestions for Authors

The manuscript thoroughly analyzes the synthesis and catabolism of the endocannabinoid system (ECS), with particular focus on N-acylethanolamines (NAEs) and anandamide (AEA) molecules. The study presents various genetic models employed to uncover the details of the ECS, including knockout models related to NAPE-PLD and GDE1 enzymes. Additionally, the paper introduces new imaging techniques such as STORM and GRABeCB2.0, and summarizes the best available genetic models in ECS research. The text is well-written in English, though there might be room for improvement in terms of word choice and sentence restructuring at some points.

The manuscript is comprehensive and well-structured. The topic is timely and may attract much attention.

I would have a few observations regarding the manuscript, which are as follows:

1. Chapter 1.1.1. Intracellular organelle-associated CB1R: Perhaps a brief explanation on how the decrease in ATP levels leads to the negative regulation of memory could be helpful.

2. Chapter 1.2. Endocannabinoids: endogenous CBR ligand: "Even though 2-AG and AEA are considered as the primary eCBs..." A brief explanation of why 2-AG and AEA are considered primary endocannabinoids could enhance clarity.

3. A brief explanation of why the "on-demand" synthesis of eCBs is unique and how it influences functional diversity in different cellular environments would be beneficial.

4. The paragraphs from 203-218 focus on the transport mechanism and hypotheses, elaborating on two possible theories for the intercellular movement of endocannabinoids (eCBs): simple diffusion and facilitated diffusion. Due to the complexity of this information, these details may be somewhat technical for readers less familiar with the topic. I suggest providing a brief summary or explanation before or during this section, introducing the concepts of simple diffusion and facilitated diffusion.

5. In the 229th line, it would be beneficial to expand or support the "heterogeneous and interconnected nature of the system" part, perhaps with examples, to further highlight the complex nature of the ECS and the challenges in research.

6. Regarding the 235th line, providing a brief explanation or example of the sentence "the best tool for a given application is dependent on several factors" could help readers better understand the factors influencing the choice of imaging tools.

7. In the 269th line, where mass spectrometry imaging is mentioned, it would be helpful to elaborate on the advantages and limitations of the method to provide readers with a more complete picture.

8. For the 269-272 lines, discussing bioengineered sensors, including a few examples or specific applications, could enhance the understanding of the significance of technological advancements for readers.

9. It would be worthwhile to highlight those model systems or methodological approaches that yielded unexpected results or encountered difficulties. These challenges and the solutions found for them can contribute to a more realistic understanding of the research process for readers. Additionally, emphasizing critical aspects such as methodological limitations or alternative interpretations of results can further strengthen the insights and contribute to the credibility of the research.

The main drawback I noticed is that the text can be overly technical and detailed at times, which may make it challenging for readers who are not fully familiar with the topic. Additionally, some parts may benefit from additional explanations or examples to ensure a clearer understanding of the subjects. For a comprehensive interpretation and overall coherence, these technical aspects and expressions need to better align with the entirety of the text.

Comments on the Quality of English Language

The text has some parts where the sentences are complex and might be hard to understand.

Author Response

We appreciate the reviewers for dedicating their time to a thorough evaluation of our manuscript and for offering constructive feedback. We are particularly pleased with their positive assessments of the importance and quality of our work. We have carefully considered their suggestions and implemented several revisions, resulting in a substantially improved version of the manuscript. Specific responses to the reviewers’ comments are provided below and the main modifications have been highlighted in yellow in the revised version of the manuscript.

We would like to point out that many of the reviewers’ concerns relate to typography (super- and subscripts in abbreviations and special characters) were an unintentional consequence of software errors in the review process. As we rely on a reference manager that is incompatible with the MDPI template, we submit the current manuscript in a plain Microsoft Word document, which the MDPI editorial team copies to the template before being sent out to reviewers. We suspect these special characters are lost in this transfer of formats, and we will do all in our power to ensure this oversight is not repeated in the following submission.

While the reviewers generally expressed satisfaction with the manuscript's English language quality, they highlighted specific cases of unclear language use. Therefore, any modifications in the manuscript not explicitly addressed in response to specific reviewer comments were made to further simplify and clarify the language.

Reviewer 4

  1. Chapter 1.1.1. Intracellular organelle-associated CB1R: Perhaps a brief explanation on how the decrease in ATP levels leads to the negative regulation of memory could be helpful.

We gratefully take the opportunity to better explain this piece of evidence for those readers who may be unfamiliar with the topic. The relevant parts of this section have now been updated to clarify the link between mtCB1R, decreased ATP and negative memory regulation.

  1. Chapter 1.2. Endocannabinoids: endogenous CBR ligand: "Even though 2-AG and AEA are considered as the primary eCBs..." A brief explanation of why 2-AG and AEA are considered primary endocannabinoids could enhance clarity.

We have now rephrased the sentence to state that 2-AG and AEA are merely the best studied, first discovered eCBs that are proven to have CBR-mediated biological functions. To avoid confusion, we avoid the term “primary” eCB, as the advancement of eCB research might disprove their unique position in this category in the future.

  1. A brief explanation of why the "on-demand" synthesis of eCBs is unique and how it influences functional diversity in different cellular environments would be beneficial.

We thank the reviewer for the opportunity to expand on the potential implications of this highly debated phenomenon. We have combined the response to this comment with that of another reviewer, requesting a clarification on the controversy related to this on-demand synthesis model. Thus, the first section referencing this phenomenon (paragraph 2, section 1.2.) has been edited to highlight the fact that eCBs seem to be the only neurotransmitters described to date that rely on this mechanism rather than the standard store-and-release model that is ubiquitous throughout the nervous system. This is what makes it unique.

Following this logic, functional diversity will thus come down to region-specific expression of enzymes that control synthesis & metabolism, and the predominant mechanisms in each area that link neuronal activation with eCB synthesis. This statement was also included but in the following section (paragraph 1, section 1.3) as it is more related to the expression of the related enzymes rather than the eCBs themselves.

  1. The paragraphs from 203-218 focus on the transport mechanism and hypotheses, elaborating on two possible theories for the intercellular movement of endocannabinoids (eCBs): simple diffusion and facilitated diffusion. Due to the complexity of this information, these details may be somewhat technical for readers less familiar with the topic. I suggest providing a brief summary or explanation before or during this section, introducing the concepts of simple diffusion and facilitated diffusion.

We agree on the complexity of this section to introductory readers, and have adapted it according to the reviewer’s helpful feedback. A brief yet thorough explanation of both simple and facilitated diffusion has now been given. The evidence for and against each theory has now been simplified into shorter, concise sentences, emphasizing in which cases the chemical nature of the eCB supports one theory or the other.

  1. In the 229th line, it would be beneficial to expand or support the "heterogeneous and interconnected nature of the system" part, perhaps with examples, to further highlight the complex nature of the ECS and the challenges in research.

We have updated this section adding new information highlighting ECS components’ multifunctionality and impact in different pathways and diseases. Two prominent examples of such crosstalk have also been added below the explanation.

  1. Regarding the 235th line, providing a brief explanation or example of the sentence "the best tool for a given application is dependent on several factors" could help readers better understand the factors influencing the choice of imaging tools.

We thank the reviewer for his/her suggestion, as we want this section to be informative to readers new to the topic as well. We have listed some factors that need to be taken in consideration when choosing the right imaging tools in this section. However, this still serves as an introduction for the further exploration of these factors in the table and text below.

  1. In the 269th line, where mass spectrometry imaging is mentioned, it would be helpful to elaborate on the advantages and limitations of the method to provide readers with a more complete picture.

We have now added some advantages and limitations in the section. This is only to introduce the reader to a more thorough description of the techniques’ benefits and downfalls in section 2.4: Imaging the cannabinoids.

  1. For the 269-272 lines, discussing bioengineered sensors, including a few examples or specific applications, could enhance the understanding of the significance of technological advancements for readers.

We have now added some advantages and limitations in the section. Again, this is only to introduce the reader to a more thorough description of bioengineered sensors’ benefits and downfalls in section 2.4: Imaging the cannabinoids.

  1. It would be worthwhile to highlight those model systems or methodological approaches that yielded unexpected results or encountered difficulties. These challenges and the solutions found for them can contribute to a more realistic understanding of the research process for readers. Additionally, emphasizing critical aspects such as methodological limitations or alternative interpretations of results can further strengthen the insights and contribute to the credibility of the research.

We thank the reviewer for this suggestion. We have now included some of current challenges that still remain to be fully addressed, as well as highlighted the existing methodological limitations. These have been added throughout the text and in “concluding remarks” in the revised version of the manuscript, and include topics such as the physiological role of mtCB1R in the CNS, the controversial presence or absence of CB2R in neurons, C. elegans as a model to study the ECS, etc.